# DNA damage independent inhibition of NF-κB transcription by anthracyclines

Angelo Ferreira Chora[1†], Dora Pedroso[2†], Eleni Kyriakou[3,4], Nadja Pejanovic[1], Henrique Colaço[2], Raffaella Gozzelino[5], André Barros[2], Katharina Willmann[2], Tiago Velho[2,6], Catarina F Moita[2], Isa Santos[2,7], Pedro Pereira[1], Silvia Carvalho[1], Filipa Batalha Martins[1], João A Ferreira[1], Sérgio Fernandes de Almeida[1], Vladimir Benes[8], Josef Anrather[9], Sebastian Weis[10,11,12], Miguel P Soares[13], Arie Geerlof[3], Jacques Neefjes[14], Michael Sattler[3,4], Ana C Messias[3,4*], Ana Neves-Costa[2*], Luis Ferreira Moita[2,15*]

[1]Instituto de Medicina Molecular, Faculdade de Medicina, Universidade de Lisboa, Lisboa, Portugal; [2]Innate Immunity and Inflammation Laboratory, Instituto Gulbenkian de Ciência, Oeiras, Portugal; [3]Institute of Structural Biology, Molecular Targets and Therapeutics Center, Helmholtz Zentrum München, Neuherberg, Germany; [4]Bavarian NMR Centre, Department of Bioscience, School of Natural Sciences, Technical University of Munich, Garching, Germany; [5]NOVA Medical School (NMS), Lisbon, Portugal; [6]Centro Hospitalar Lisboa Norte - Hospital de Santa Maria, EPE, Avenida Professor Egas Moniz, Lisbon, Portugal; [7]Serviço de Cirurgia, Centro Hospitalar de Setúbal, Setúbal, Portugal; [8]EMBL Genomics Core Facilities, Heidelberg, Germany; [9]Feil Family Brain and Mind Research Institute, Weill Cornell Medicine, New York, United States; [10]Institute for Infectious Disease and Infection Control, Friedrich-Schiller University, Jena, Germany; [11]Department of Anesthesiology and Intensive Care Medicine, Jena University Hospital, Friedrich-Schiller University, Jena, Germany; [12]Leibniz Institute for Natural Product Research and Infection Biology, Hans Knöll Institute (HKI), Jena, Germany; [13]Inflammation Laboratory, Instituto Gulbenkian de Ciência, Oeiras, Portugal; [14]Department of Cell and Chemical Biology, LUMC, Leiden, Netherlands; [15]Instituto de Histologia e Biologia do Desenvolvimento, Faculdade de Medicina da Universidade de Lisboa, Lisbon, Portugal

*For correspondence:
ana.messias@helmholtz-muenchen.de (ACM);
arcosta@igc.gulbenkian.pt (AN-C);
lferreiramoita@gmail.com (LFM)

[†]These authors contributed equally to this work

**Abstract** Anthracyclines are among the most used and effective anticancer drugs. Their activity has been attributed to DNA double-strand breaks resulting from topoisomerase II poisoning and to eviction of histones from select sites in the genome. Here, we show that the extensively used anthracyclines Doxorubicin, Daunorubicin, and Epirubicin decrease the transcription of nuclear factor kappa B (NF-κB)-dependent gene targets, but not interferon-responsive genes in primary mouse (*Mus musculus*) macrophages. Using an NMR-based structural approach, we demonstrate that anthracyclines disturb the complexes formed between the NF-κB subunit RelA and its DNA-binding sites. The anthracycline variants Aclarubicin, Doxorubicinone, and the newly developed Dimethyl-doxorubicin, which share anticancer properties with the other anthracyclines but do not induce DNA damage, also suppressed inflammation, thus uncoupling DNA damage from the effects on inflammation. These findings have implications for anticancer therapy and for the development of novel anti-inflammatory drugs with limited side effects for life-threatening conditions such as sepsis.

## Editor's evaluation

This is an interesting study with several implications for anti-cancer therapy, both systemic and local. This study contributes additional very important cancer cell-specific NF-κB inhibition information which is required for improving anticancer efficacy and reducing systemic toxicity. The conceptual significance is that anthracyclines prevent the induction of pro-inflammatory genes in macrophages without inducing or involving DNA damage response. Confirmation of this mechanism in vivo will help the future development of more efficacious anthracycline drugs specifically optimized for this mechanism without causing major side effects.

## Introduction

In innate immune cells, including macrophages, the recognition of microbial-associated molecular patterns and non-microbial molecules by specialized germline-encoded pattern recognition receptors activates the transcription of a large number of pro-inflammatory genes (*Medzhitov and Horng, 2009*). The nuclear factor kappa B (NF-κB) family of transcription factors, comprising p65/RelA, RelB, c-Rel, p100, and p105, plays a prominent role in regulating inflammatory gene induction while preventing cytotoxicity (*Hayden and Ghosh, 2008*).

The most abundant member of the NF-κB family, RelA, is retained in the cytoplasm by the inhibitor protein IκBα. In response to a variety of stimuli, IκBα phosphorylation and subsequent proteasomal degradation frees RelA for nuclear translocation and DNA binding (*Chen and Greene, 2004*; *Ghosh and Baltimore, 1990*). RelA-dependent IκBα re-synthesis is required for RelA nuclear eviction and the timely termination of NF-κB-dependent transcription (*Sun et al., 1993*; *Beg and Baldwin, 1993*). NF-κB sustained activity can perpetuate inflammation and contribute to the pathogenesis of many chronic conditions (*Baldwin, 2001*).

Anthracyclines, mostly Doxorubicin, have been used for many decades against a wide variety of cancers (reviewed in *Hande, 1998*). These anticancer drugs target topoisomerase II (TopoII), thereby inducing DNA damage, including DNA double-strand breaks (*Tewey et al., 1984*). The stabilization of the otherwise transient TopoII–DNA complex by anthracyclines activates DNA damage responses (DDRs) and ultimately senescence or programmed cell death by apoptosis (*Nitiss, 2009*; *Eom et al., 2005*). Other biological activities of anthracyclines with potential clinical impact include DNA intercalation, helicase inhibition, and free radical formation (*Hande, 1998*). More recently, anthracyclines were shown to evict histones from discrete chromosomal regions and to contribute to apoptosis in a TopoII-independent manner (*Pang et al., 2013*; *Yang et al., 2013*).

We have previously shown that anthracyclines induce disease tolerance to infection in vivo. In mouse models of sepsis, low doses of anthracyclines, such as Epirubicin, led to less severe disease and decreased mortality independently of circulating and organ pathogen load (*Figueiredo et al., 2013*). We further found that ATM (Ataxia Telangiectasia Mutated), a master DNA damage sensor (*Maréchal and Zou, 2013*), is required for Epirubicin-mediated disease tolerance (*Figueiredo et al., 2013*). In breast cancer cells, Epirubicin was already known to activate ATM (*Millour et al., 2011*). In addition to roles in DDRs, ATM participates in a complex network of signaling pathways that intersect with NF-κB (*Huang et al., 2010*; *Piret et al., 1999*; *Li et al., 2001*; *Wu et al., 2006*). One central finding in Epirubicin-treated septic animals, which we also observed in the monocytic cell line THP-1 challenged with pro-inflammatory stimuli, was the strong suppression of cytokine secretion (*Figueiredo et al., 2013*). The mechanism whereby anthracyclines limit cytokine production remains unstudied (*Neves-Costa and Moita, 2017*) but a recent report has reproduced this finding in human macrophages (*Köse-Vogel et al., 2019*). Here, we investigate how low doses of anthracyclines regulate the pro-inflammatory transcriptional program in primary mouse macrophages and ask whether the DNA damaging activities of anthracyclines are required for the transcriptional downregulation of cytokines and other pro-inflammatory mediators.

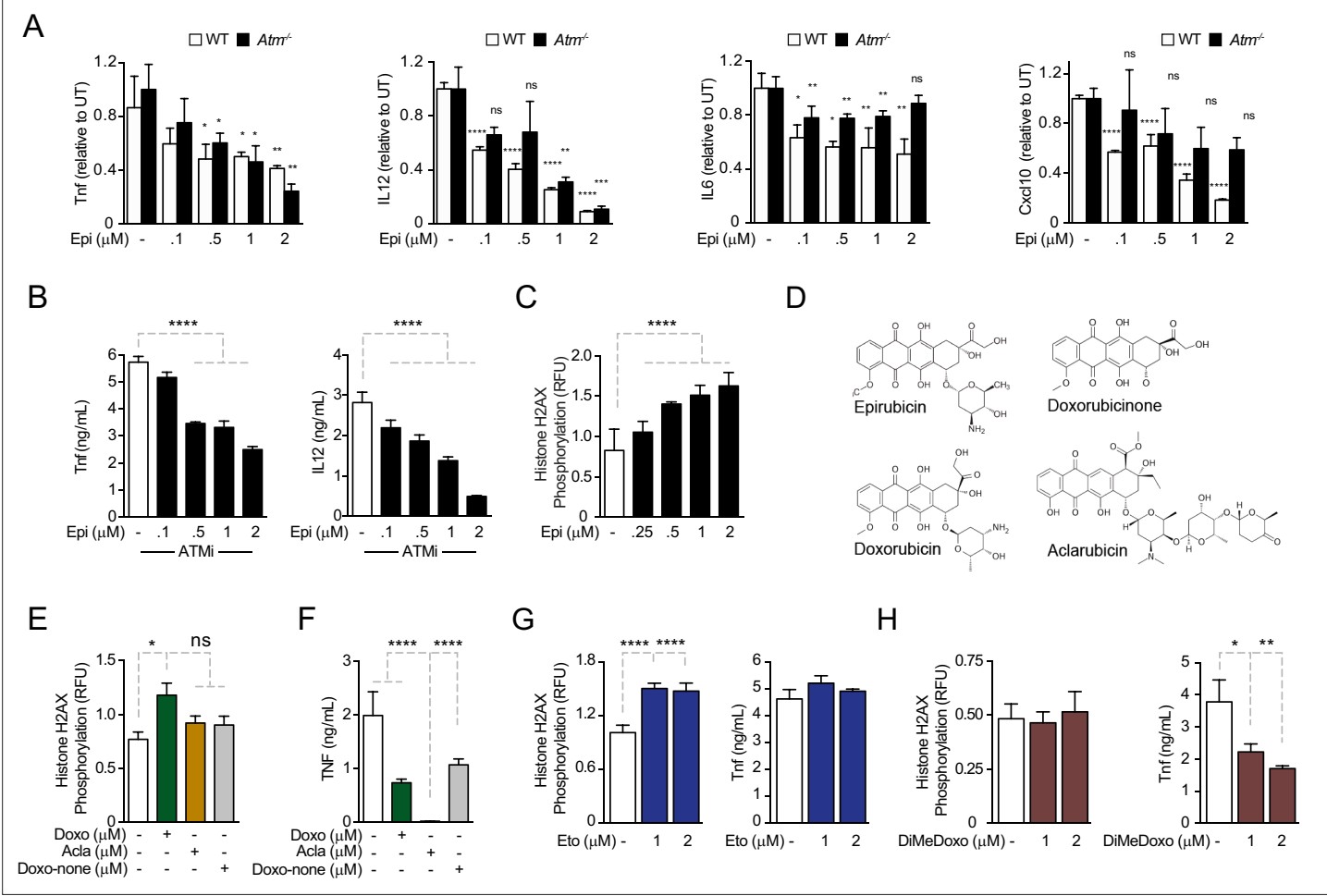

**Figure 1.** Anthracyclines regulate cytokine secretion independently of ATM in bone marrow-derived macrophages following *E. coli* challenge. (**A**) Cytokine secretion of TNF, IL12, IL6, and Cxcl10 was quantified by ELISA following *E. coli* challenge in the presence of various doses of Epirubicin (Epi) in WT and *Atm*$^{-/-}$ macrophages. (**B**) Secretion of TNF and IL12 was quantified in macrophages treated with various doses of Epi and the ATM inhibitor KU-55933. (**C**) H2AX phosphorylation was quantified by ELISA in Epi-treated macrophages, normalized to total H2AX and shown as relative fluorescence intensities (RFU). (**D**) Schematic representation of the molecular structures of the anthracyclines Epi, Aclarubicin (Acla), Doxorubicin (Doxo), and Doxorubicinone (Doxo-none). (**E**) H2AX phosphorylation was quantified in the presence of Doxo, Acla, and Doxo-none. (**F**) TNF secretion was quantified in the presence of Doxo, Acla, and Doxo-none. (**G, H**) H2AX phosphorylation and TNF secretion were quantified in macrophages treated with Etoposide (Eto) and Dimethyl-doxorubicin (diMe-Doxo) following *E. coli* challenge. The assays show arithmetic means and standard deviations of technical replicates from one representative animal of at least three independent animals tested. *p < 0.05; **p < 0.01; ***p < 0.001; ****p < 0.0001.

The online version of this article includes the following figure supplement(s) for figure 1:

**Figure supplement 1.** Control experiments for cytokine regulation by Epirubicin (Epi) and Aclarubicin (Acla) and characterization of DNA damage.

**Figure supplement 2.** Similar to Epirubicin (Epi), Aclarubicin (Acla) regulates cytokine secretion independently of ATM.

## Results

### Cytokine secretion and DNA damage

We investigated the role of DDRs in cytokine downregulation induced by anthracyclines (*Figueiredo et al., 2013*). We started by measuring cytokine concentrations in conditioned media in response to Epirubicin (Epi) upon activation of WT and *Atm*-deficient bone marrow-derived mouse macrophages (BMDMs) with pro-inflammatory stimuli. We observed that WT and *Atm*$^{-/-}$ BMDMs secreted comparable amounts of inflammatory cytokines (*Figure 1—figure supplement 1A*). Pre-treatment of BMDMs with Epi downregulated cytokines following *E. coli* challenge independently of ATM, with Epi leading to a dose-dependent decrease in TNF (Tumor Necrosis Factor), IL12, IL6, and Cxcl10 production not only in WT, but also in *Atm*$^{-/-}$ macrophages (*Figure 1A*). Sporadically, *Atm*$^{-/-}$ cells may

show different levels of cytokine secretion relative to WT for some Epi doses, but the difference is small in magnitude (**Figure 1A**). We then tested cytokine secretion in the presence of the ATM inhibitor Ku-55933 and again observed that ATM is dispensable for the effect of Epi in limiting TNF and IL12 secretion (**Figure 1B**). Cytotoxicity due to Epi treatments was not observed at the doses used (**Figure 1—figure supplement 1B**).

As cytokine downregulation by Epi was independent of ATM, we hypothesized that the DDR was not required for the effects of Epi on cytokine suppression. Using the alkaline comet assay to detect strand breaks, we observed that DNA damage caused by Epi was time- and dose dependent (**Figure 1—figure supplement 1C**). We then quantified the DNA damage using phosphorylation of histone H2AX at Ser139 (γH2AX) as a surrogate marker. In BMDMs, DNA damage was strongly induced by Epi in the range of concentrations that modulated cytokine production (**Figure 1C**) and was comparable to that of Etoposide (Eto), another well-studied TopoII inhibitor that causes DNA breaks and induces ATM-mediated DDRs (**Figure 1—figure supplement 1D**; **Caporossi et al., 1993**; **Banáth and Olive, 2003**). γH2AX levels also showed that the inflammatory challenge did not induce significant extra DNA damage in the conditions used (**Figure 1—figure supplement 1E**). The lack of interdependency between DNA damage and cytokine modulation was then assessed for other anthracyclines. These drugs all share a tetracycline ring decorated with one or multiple amino sugars (**Figure 1D**) and their activities range from failing to induce DNA breaks to being potent inducers of double-strand breaks (**Pang et al., 2013**). Eto is not a member of the anthracycline class but inhibits TopoII like Doxo and Epi. We quantified the DNA damage induced by the different anthracyclines in BMDMs and observed that whereas Epi and the closely related Doxorubicin (Doxo) led to a dose-dependent increase inγH2AX signal, Aclarubicin (Acla) and Doxorubicinone (Doxo-none) did not induce significant damage (**Figure 1C, E**). The comet assay further established that DNA damage caused by Acla is comparable to the basal damage in untreated cells, in sharp contrast with the highly damaging effects of Epi and Eto (**Figure 1—figure supplement 1F**). Acla and Doxo-none were then tested for their ability to regulate cytokines. Acla was very strong at downregulating all the cytokines tested at concentrations without significant toxicity (**Figure 1—figure supplement 2A–E**). Whereas downregulation of TNF and IL12 was a common property of the anthracyclines tested, Doxo-none was less potent than Epi, Doxo, and Acla (**Figure 1F** and compare with **Figure 1A**). Doxo-none differs from the other anthracyclines for not inducing DNA breaks nor histone eviction (**Qiao et al., 2020**). To further uncouple the DDR from cytokine modulation by anthracyclines we tested whether Eto and a variant of Doxo, Dimethyl-doxorubicin (diMe-Doxo), could also downregulate cytokines. diMe-Doxo was made incapable of causing DNA damage while leaving histone eviction (**Qiao et al., 2020**). **Figure 1G, H** shows that TNF was not downregulated by Eto but was by diMe-Doxo. In line with this observation, none of the anthracyclines tested regulated cytokine production differently in $Atm^{-/-}$ BMDMs or in the presence of the ATM inhibitor (**Figure 1—figure supplement 2F–I**). Therefore, structural damage to DNA alone is unlikely to explain the observed effect of anthracyclines on cytokine secretion. Instead, the control of cytokine secretion upon inflammatory challenges is likely to be more associated with the histone eviction activity of the anthracyclines.

## Anthracyclines negatively regulate the transcription of NF-κB target genes

Before dissecting the molecular mechanisms of cytokine downregulation by anthracyclines, we investigated the in vivo effects of the administration of these drugs in a model of septic shock. To this end, we co-administered Epi or Acla to mice that were challenged with *E. coli*. We found that the circulating levels of TNF were significantly decreased in mice treated with either Epi or Acla for 8 hr following the initial challenge (**Figure 2A**), suggesting physiological relevance for our findings. We then investigated the effects of anthracyclines on transcription and their dependence on DNA damage. We performed RNA sequencing (RNA-seq) in BMDMs pre-treated with either Epi or Acla and challenged thereafter with lipopolysaccharide (LPS; **Figure 2—figure supplement 1A**). By comparing mRNA levels of untreated with those of 4 hr LPS-treated BMDMs, we detected strong induction of pro-inflammatory gene expression, in line with the well-described patterns of transcriptional regulation in response to this TLR4 agonist (**Figure 2B**, lane 1). Differential expression analysis revealed 455 genes at least fivefold upregulated in LPS-treated BMDMs compared with untreated, whereas only 75 genes were downregulated to the same extent (**Figure 2—figure supplement 1B**). The transcriptomes also

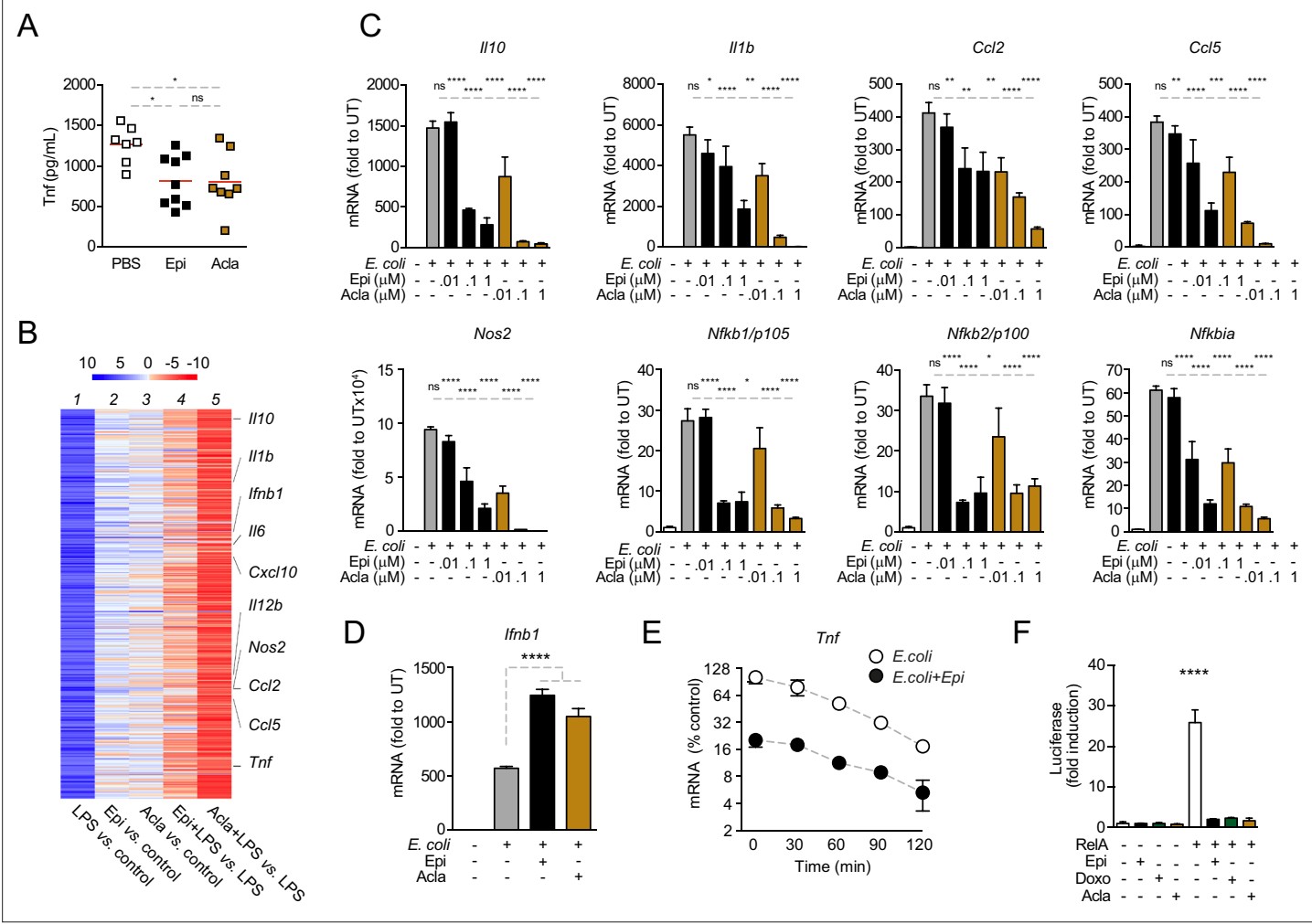

**Figure 2.** Anthracyclines lower circulating TNF levels in an in vivo model of sepsis and downregulate nuclear factor kappa B (NF-$\kappa$B)-dependent transcription. (**A**) TNF concentrations in serum 8 hr post *E. coli* challenge in mice treated with phosphate-buffered saline (PBS, *n* = 7), Epirubicin (Epi, *n* = 9), or Aclarubicin (Acla, *n* = 8). (**B**) RNA sequencing (RNA-seq) in macrophages stimulated with lipopolysaccharide (LPS) for 4 hr and treated with Epi or Acla. (**C**) Gene expression was quantified by quantitative RT-PCR in macrophages following *E. coli* challenge and treated with various doses of Epi and Acla; results were normalized to expression in untreated conditions (UT). (**D**) Gene expression of *Ifnb1* was quantified as in C. (**E**) Analysis of *Tnf* mRNA stability in macrophages treated with 2 μM of Epi for 1 hr or left untreated; Actinomycin D was added 2 hr after *E. coli* stimulation. (**F**) Luciferase quantification of a $\kappa$B reporter in HEK293 cells treated with 2 μM of Epi, Doxo, and Acla in the absence or presence of a vector overexpressing RelA. C–E show arithmetic means and standard deviations of technical replicates from one representative animal of at least three independent animals tested. F shows one representative experiment of two independently performed assays. *p < 0.05; **p < 0.01; ***p < 0.001; ****p < 0.0001.

The online version of this article includes the following source data and figure supplement(s) for figure 2:

**Source data 1.** Functional analysis of the RNA sequencing (RNA-seq) data.

**Figure supplement 1.** Testing specificity of transcriptional regulation by anthracyclines.

**Figure supplement 2.** CRISPR/Cas9-mediated knock-out (KO) of RelA in HEK293 cells abolishes the response to TNF.

**Figure supplement 2—source data 1.** Western blot in *Figure 2—figure supplement 2*.

**Figure supplement 2—source data 2.** Original blot in *Figure 2—figure supplement 2*.

**Figure supplement 2—source data 3.** Molecular weight marker in original blot in *Figure 2—figure supplement 2*.

pointed to specific gene expression signatures by Epi and Acla, partly overlapping but not identical, both in unstimulated BMDMs (*Figure 2B*, lanes 2 and 3) and in the presence of LPS (*Figure 2B*, lanes 4 and 5). This could be due to different genomic location preferences between these drugs, as shown for Acla versus Daunorubicin (Dauno; *Pang et al., 2015*). As anticipated from the cytokine secretion results, repression of target genes by Acla was stronger than by Epi (*Figure 2B*, compare lanes 4

and 5). Functional enrichment analysis showed that both Epi and Acla repress cytokine production, among other effector functions of macrophages (*Figure 2—source data 1*). The fact that both Epi and Acla regulate inflammatory gene expression also supports a DNA damage-independent mechanism. Quantitative RT-PCR corroborated the downregulation of *Tnf*, *Il12*, *Il6*, and *Cxcl10* by both Epi and Acla (*Figure 2—figure supplement 1C*), and also of a broader subset of pro-inflammatory mediators, including *Nfkbia*, the gene encoding the RelA inhibitor IκBα (*Figure 2C*). We then searched for promoter motifs in repressed genes (*Chang and Nevins, 2006*) and, not surprisingly, we detected an NF-κB signature shared by genes mutually downregulated by Epi and Acla. We extended the RT-PCR analysis to other TLR4-induced transcriptional programs, namely IRF-dependent interferon transcription (*Kawai and Akira, 2007*). IRF3-dependent *Ifnb1* transcription was strongly induced by *E. coli* as expected, but not downregulated by Epi or Acla (*Figure 2D*). In agreement, IFNβ secretion was also not suppressed by the anthracyclines tested (*Figure 2—figure supplement 1D*). *Ifna1* and *Ifna4* transcription, typically mediated by virally induced receptors such as TLRs7/9, was not induced, nor was the expression of these genes further downregulated by Epi or Acla (*Figure 2—figure supplement 1E*). In combination, these results indicate that anthracyclines do not compromise the overall cellular transcription. Instead, the transcriptional profiles point to a negative effect of Epi and Acla specifically on NF-κB-regulated gene expression. This is compatible with previous observations that the anthracyclines Doxo and Dauno repress TNF-induced NF-κB transactivation in cancer cells (*Campbell et al., 2004*; *Ho et al., 2005*).

To test whether anthracyclines affect mRNA stability of pro-inflammatory genes, we treated BMDMs with actinomycin D (ActD) 2 hr after *E. coli* stimulation in the presence or absence of Epi and measured the rate of mRNA decay. We did not observe significant differences in the half-lives of *Tnf* mRNAs between non-treated and Epi-exposed BMDMs (*Figure 2E*). The effect of anthracyclines on NF-κB-dependent gene expression is therefore likely to be caused by changes in transcription and not due to regulation of mRNA stability.

To further address transcriptional regulation by anthracyclines, we used a κB luciferase reporter assay in HEK293 cells. We started by testing whether RelA is required for reporter induction in these cells following inflammatory stimulation and for that we decided to knock out RelA (*Figure 2—figure supplement 2A, B*). Not surprisingly, RelA KO compromised cell viability and therefore these cells were only used immediately after CRISPR/Cas9 editing (*Figure 2—figure supplement 2C*). As expected, reporter expression was induced several times in CRISPR control cells after stimulation with TNF (*Figure 2—figure supplement 2D*). However, the expression in RelA KO was very similar to the basal expression of the reporter without any stimulus (*Figure 2—figure supplement 2D*), which suggested that RelA is critical for luciferase expression. We also tested the requirement of RelA for endogenous gene expression and observed that, similar to the reporter, induction of the TNF gene is largely dependent on RelA (*Figure 2—figure supplement 2E*). Our observations are compatible with RelA being the main factor required for NF-κB-mediated pro-inflammatory transcription and therefore we tested whether Epi, Doxo, and Acla were able to regulate reporter expression induced by RelA overexpression. We co-transfected the reporter together with a vector expressing full-length RelA and observed considerable induction of luciferase (*Figure 2F*), which was markedly inhibited by the anthracyclines Epi, Doxo, and Acla (*Figure 2F*). From our observations, anthracyclines are expected to target RelA directly or indirectly.

## Epirubicin affects NF-κB subcellular localization

NF-κB nuclear translocation is required for DNA binding and initiation of transcription (*Ghosh and Baltimore, 1990*). We therefore asked whether the downregulation of NF-κB targets by anthracyclines was due to impaired RelA nuclear translocation. We observed that RelA nuclear levels in untreated BMDMs rapidly increased following inflammatory challenge and RelA slowly relocated overtime, being mostly cytoplasmic 4 h after stimulation (*Figure 3A*, control panel). In BMDMs exposed to Epi, RelA translocated to the nucleus upon *E. coli* challenge, but despite decreased NF-κB-dependent gene expression, RelA remained nuclear at all time points analyzed (*Figure 3A*, Epi panel). Proteolytic degradation and re-synthesis of the NF-κB inhibitor IκBα are a central mechanism controlling the subcellular localization of NF-κB factors (*Ghosh and Baltimore, 1990*; *Sun et al., 1993*; *Beg and Baldwin, 1993*). From our mRNA analysis, we expected Epi to affect IκBα (*Figure 2C*), and therefore we tested the effects of Epi on IκBα protein throughout time. As extensively reported, IκBα was

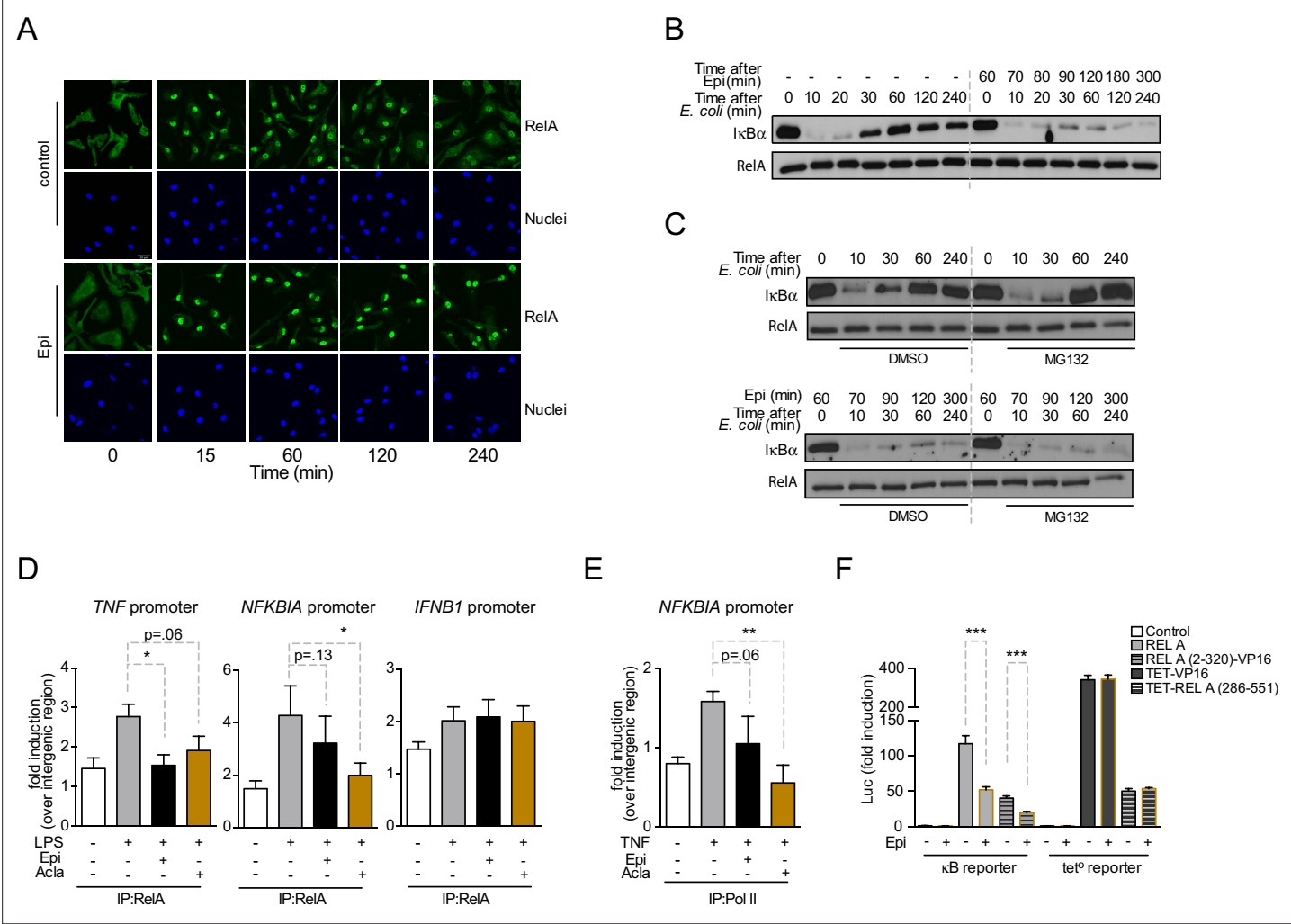

**Figure 3.** Anthracyclines affect RelA subcellular localization and compromise RelA binding to *NFKBIA* and *TNF* promoters, but not to *IFNB1* promoter. Reporter assays suggest that the RelA domain targeted by Epirubicin is the REL-homology domain (RHD). (**A**) RelA immunolocalization in macrophages challenged with *E. coli* for 4 hr and left untreated (control) or treated with 2 μM of Epirubicin (Epi); scale bar = 20 μm. (**B**) IκBα degradation kinetics in macrophages following *E. coli* challenge in the absence or presence of 2 μM of Epi for 1 hr at the time of *E. coli* challenge and using total RelA levels as loading control. (**C**) Macrophages were either left untreated (top panel) or treated with Epi (bottom panel) and challenged with *E. coli* for the indicated times in the presence of the proteasome inhibitor MG132 (10 μM) or its vehicle DMSO; IκBα degradation was assessed using total RelA as loading control. (**D**) Macrophages were challenged with lipopolysaccharide (LPS), treated with 2 μM of Epi or Aclarubicin (Acla) and an anti-RelA antibody was used to immunoprecipitate the associated chromatin, from where the promoter sequences of *NFKBIA*, *TNF*, and *IFNB1* were amplified. (**E**) HEK293 cells were challenged with TNF, treated with 2 μM of Epi or Acla and an anti-PolII antibody was used to immunoprecipitate the associated chromatin, from where the promoter sequence of *NFKBIA* promoter was amplified. (**F**) HEK293 cells were transiently transfected with a κB-luciferase reporter alone or in conjunction with full-length RelA or RelA (2-320)-VP16 or TET-RelA (268-551) and cells were left untreated or treated with 2 μM of Epi for 16 hr. A–C show one representative experiment using macrophages from at least three independent animals tested; D and E show arithmetic means and standard deviations of technical replicates from one representative experiment of at least four independently performed assays; F shows arithmetic means and standard deviations of technical replicates from one representative experiment of two independently performed assays. *p < 0.05; **p < 0.01; ***p < 0.001.

The online version of this article includes the following source data and figure supplement(s) for figure 3:

**Source data 1.** Western blot in *Figure 3B*.

**Source data 2.** Original blot of IκBα in *Figure 3B*.

**Source data 3.** Original blot of total RelA in *Figure 3B*.

**Source data 4.** Western blot in *Figure 3C*.

**Source data 5.** Original blot of IκBα in *Figure 3C*, no Epirubicin (upper panel, no Epi).

**Source data 6.** Original blot of IκBα in *Figure 3C*, with Epirubicin (lower panel, Epi).

*Figure 3 continued on next page*

*Figure 3 continued*

**Source data 7.** Original blots of total RelA in *Figure 3C* (both panels).

**Figure supplement 1.** Epirubicin (Epi) modulates nuclear factor kappa B (NF-$\kappa$B) activity in HEK293 cells.

degraded and protein levels restored to initial values within 60 min in non-treated BMDM. In contrast, Epi pre-treatment profoundly diminished I$\kappa$B$\alpha$ cellular levels throughout the time course (*Figure 3B*), as anticipated from our previous results. Reduced stimulus-induced I$\kappa$B$\alpha$ synthesis had already been reported to promote nuclear localization of RelA, an effect not associated with increased NF-$\kappa$B transcriptional activity (*Hochrainer et al., 2007*).

A role for Doxo in proteasome activation has been proposed (*Liu et al., 2008*). To test if increased proteasome-dependent degradation by anthracyclines also contributes to low I$\kappa$B$\alpha$ levels, we examined I$\kappa$B$\alpha$ degradation kinetics upon inhibition of the proteasome. While blockage of proteasome activity by MG-132 increased I$\kappa$B$\alpha$ protein levels upon de novo synthesis following *E. coli* challenge (*Figure 3C*, upper panel), I$\kappa$B$\alpha$in cells pre-exposed to Epi never recovered (*Figure 3C*, lower panel). It is therefore unlikely that I$\kappa$B$\alpha$ regulation by Epi is due to any role for the anthracyclines in proteasomal activity. Our data suggest that anthracyclines are capable of breaking the critical negative feedback loop that maintains the cellular responsiveness to subsequent inflammatory stimuli due to their role in repressing *NFKBIA* mRNA expression.

## Epirubicin and Aclarubicin regulate NF-$\kappa$B binding to its targets

We further investigated by chromatin immunoprecipitation (ChIP) how anthracyclines suppress NF-$\kappa$B-dependent pro-inflammatory programs. Firstly, we established that the effects of anthracyclines in BMDMs could be replicated in HEK293 cells without affecting cell viability (*Figure 3—figure supplement 1A–C*). ChIP was then performed at the promoter of selected NF-$\kappa$B target genes in HEK293 cells. We observed that RelA binding following inflammatory activation was weakened by pre-treatment with Epi and Acla, ranging from a small reduction in binding to statistically significant impairments (*Figure 3D*, *TNF* and *NFKBIA* promoters). In contrast, binding of RelA to the *IFNB1* promoter was not affected (*Figure 3D*, right). Recruitment of RNA PolII to promoters of NF-$\kappa$B target genes, including *NFKBIA*, was also compromised by Epi and in a more pronounced and statistically significant way, by Acla (*Figure 3E*). These results are consistent with previous observations that Doxo treatment strongly reduced the association between RelA and DNA, which was proposed to be a consequence of defective post-transcriptional modifications in NF-$\kappa$B subunits (*Ho et al., 2005*). Furthermore, Doxo is also responsible for the reduced recruitment of at least one other transcription factor, HIF-1$\alpha$, to its targets (*Tanaka et al., 2012*).

RelA consists of a REL-homology domain (RHD) and a transactivation domain (TAD, *Figure 3—figure supplement 1D*). The RHD includes the DNA-binding domain (DBD), the dimerization domain (DD), and the nuclear localization sequence (NLS) (*Toledano et al., 1993*). We asked which RelA domain is targeted by Epi. For that, two different chimeric DNA constructs were tested: (1) RelA RHD fused to a TAD derived from the Herpes simplex virus VP16 protein (RelA (2-320)-VP16); and (2) DBD from the bacterial tetracycline repressor (TET) fused to the RelA TAD (TET-RelA (286-551)) (*Anrather et al., 1999*). As observed for full-length RelA (*Figure 2F*), Epi significantly inhibited the transcriptional activity of RelA (2-320)-VP16, as quantified in HEK293 cells transiently co-transfected with the $\kappa$B luciferase reporter (*Figure 3F*). However, Epi failed to inhibit the transcriptional activity of both TET-RelA (286-551) and TET fused to VP16 (TET-VP16) construct, as observed using a tetracycline operon luciferase (tet°-luc) reporter (*Figure 3F*). These results suggest that Epi compromises RelA transcriptional activity by targeting the RHD. Since Epi does not affect the TAD, this result points to a mechanism of action different from NF-$\kappa$B regulation by Doxo and Dauno in cancer cells (*Campbell et al., 2004*).

## Epirubicin and Aclarubicin bind to a $\kappa$B-33 promoter sequence

To understand how anthracyclines affect binding of NF-$\kappa$B to its targets, we used NMR and biophysical experiments. We tested a 14-mer DNA sequence (5'-CTGGAAATTTCCAG-3') derived from the NF-$\kappa$B-33 promoter (*Chen et al., 2000*; *Chen et al., 1998*). As expected, we detected binding of Epi and Acla to the DNA duplex by NMR spectroscopy, with dramatic changes in the imino region

of the DNA after the addition of the drugs (*Figure 4A, B* [black and red spectra], region 11–14 ppm) suggesting that the anthracyclines disturb the Watson–Crick hydrogen bonds in the double helix. Using isothermal titration calorimetry (ITC), we determined that two molecules of Epi or Acla bind to the DNA duplex with equilibrium dissociation constants ($K_D$) of 11.4 and 11.7 µM, respectively (*Table 1* and *Figure 4—figure supplement 1A, B*), indicating that Acla and Epi bind to the κB-33-derived DNA sequence with similar affinity. The affinity of Acla to different DNA molecules has been reported to be in the low nM to low µM range (*Furusawa et al., 2016*; *Skovsgaard, 1987*; *Utsuno and Tsuboi, 1997*), while Doxo, which differs from Epi in the stereochemistry of the 4′-OH group, binds to DNA in the nM range (*Katenkamp et al., 1983*). Most of the anthracyclines have a preference for GC- or TG-rich sequences, with the aglycon chromophore of the anthracyclines intercalating at a pyrimidine–purine step, and the sugar part interacting with the DNA minor groove (*Chaires, 2015*; *Frederick et al., 1990*; *Temperini et al., 2003*). The NF-κB-33 DNA oligonucleotide used in our assays contained two 5′-TG-3′ motifs suggesting that two molecules of anthracyclines can bind to it. When we used size-exclusion chromatography (SEC) coupled with static light scattering (SLS) of RelA incubated with DNA and excess anthracyclines, an increase in the molecular weight (MW) of the DNA was observed, further pointing to binding of anthracyclines to DNA. However, due to their small MW, it was difficult to establish an accurate stoichiometry, and ITC proved to be more reliable, as discussed above (*Figure 4—figure supplement 2* and *Figure 4—source data 1*).

## RelA RHD binds tightly to NF-κB-33

The crystal structure of RelA RHD in complex with a 18-mer DNA segment derived from the κB-33 promoter revealed that the double-stranded DNA mainly contacts the interface between the DD and the DBD, with one of the protein monomers recognizing the 5′-GGAA-3′ site, and the other the 5′-GAAA-3′ site (*Figure 4—figure supplement 3A*; *Chen et al., 1998*). Using NMR spectroscopy, we confirmed binding of the 14-mer DNA to RelA RHD containing the DD and DBD domains (residues 19–291, *Figure 4C, D* [black and red spectra], *Figure 4—figure supplement 3B*). As expected, the DNA thymine and guanine iminos remained detectable upon RelA addition but underwent line broadening consistent with the DNA maintaining its double helix structure upon protein binding (*Figure 4C, D* [black and red spectra]); at the same time, addition of DNA to RelA caused chemical shift perturbations and line broadening of backbone amides on the DNA-binding site (*Figure 4—figure supplement 3B*). In addition, titration of DNA to RelA by ITC revealed strong binding, with an affinity of 230 nM (*Table 1* and *Figure 4—figure supplement 1C*). This value is very similar to the reported value of 256 nM obtained with a murine RHD construct and a 18-mer DNA using fluorescence polarization (*Chen et al., 2000*). Notably, the DNA–protein interaction is endothermic, that is enthalpically unfavorable and thus entropy driven. By combining SEC and SLS experiments, we confirmed that both the protein and the DNA form dimers (*Figure 4—figure supplement 2A, B* and *Figure 4—source data 1*) but protein–DNA complexes were not observed, possibly due to a very fast off-rate (*Bergqvist et al., 2009*).

## Epirubicin and Aclarubicin disturb RelA-κB-33 binding

We then studied the effect of Epi and Acla on RelA–DNA binding by NMR, by following the effect either on the DNA or on the protein. Binding of RelA to the DNA did not substantially affect the DNA double helix as judged by DNA imino protons in thymine and guanine bases, as their intensity and chemical shifts were not substantially changed (*Figure 4C, D*; black and red spectra). However, subsequent addition of anthracyclines clearly altered the Watson–Crick base-paring even when bound to RelA (*Figure 4C, D*; blue spectra). In contrast, titration of DNA with anthracyclines caused severe line broadening of the DNA base imino groups, which was maintained upon RelA addition (*Figure 4A, B*). As the anthracyclines are expected to bind to the TG base pairs in the κB sequence located at the ends of the double helix (*Figure 5A*), the DNA helix, though distorted, is still able to bind to RelA.

Protein-based titration experiments to determine how RelA binding to DNA is affected by anthracyclines were performed in two ways: (1) RelA was first titrated with DNA and then with anthracyclines and (2) RelA was initially titrated with anthracyclines and then with DNA. Addition of anthracyclines to RelA titrated with DNA caused chemical shift perturbations and line-broadening beyond detection of some backbone resonances consistent with Epi and Acla perturbing the RelA–DNA complex but not disrupting it (*Figure 5B*, compare green and red spectra) as the final spectra (green) do not

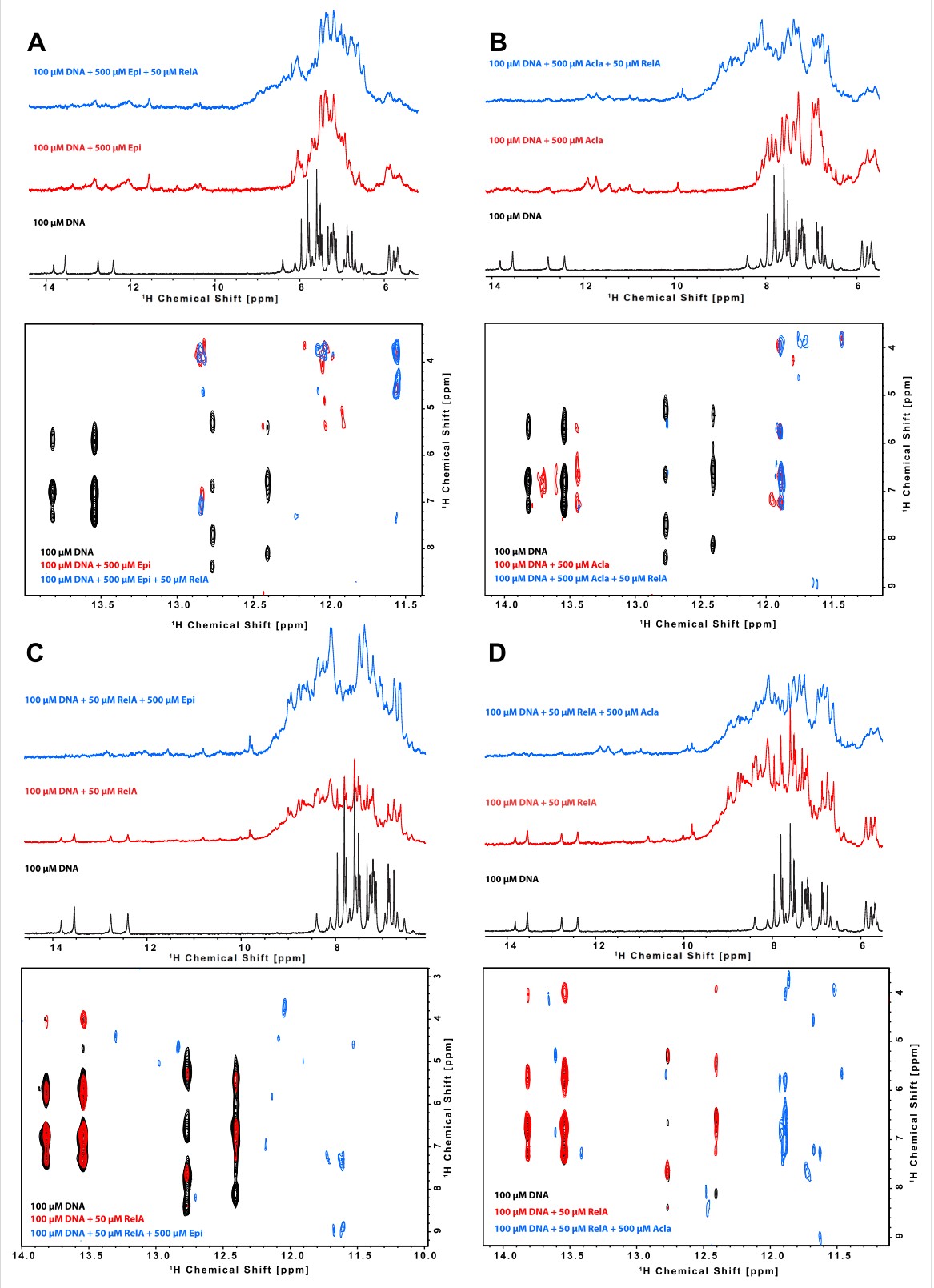

**Figure 4.** Characterization of the binding of anthracyclines to NF-$\kappa$B-33 promoter DNA sequence and their effect on RelA–DNA complex as assessed by following the DNA imino protons by NMR. 1D $^1$H spectrum (top) and zoom of the cross peaks of the imino protons with the deoxyribose in a 2D $^1$H, $^1$H-NOESY (bottom) of a 100 µM 14-mer duplex DNA solution and subsequent addition of (**A**) 500 µM Epirubicin (Epi, red) and 50 µM $^2$H, $^{15}$N-RelA$_{19-291}$ dimer (blue), (**B**) 500 µM Aclarubicin (Acla, red) and 50 µM $^2$H, $^{15}$N-RelA$_{19-291}$ dimer (blue), (**C**) 50 µM $^2$H, $^{15}$N-RelA$_{19-291}$ dimer (red) and 500 µM Epi (blue),

*Figure 4 continued on next page*

*Figure 4 continued*

and (**D**) $^2$H, $^{15}$N-RelA$_{19-291}$ dimer (red) and 500 µM Acla (blue). Experiments were recorded at 800 MHz and 10°C in 100 mM d11-Tris–HCl pH 7.5, 75 mM NaCl, 5 mM d10-DTT, and 10% D$_2$O.

The online version of this article includes the following source data and figure supplement(s) for figure 4:

**Source data 1.** Molecular weight values of RelA$_{19-291}$, $\kappa$ B-33 DNA, $\kappa$ B-33 DNA + Epirubicin (Epi), and $\kappa$ B-33 DNA + Aclarubicin (Acla) using SEC in combination with static light scattering (SLS).

**Figure supplement 1.** Isothermal titration calorimetry (ITC)-binding measurements.

**Figure supplement 2.** Determination of the molecular weight (MW) of complexes using size-exclusion chromatography (SEC) in combination with static light scattering (SLS).

**Figure supplement 3.** RelA REL-homology domain (RHD) binding to $\kappa$ B-33 promoter DNA.

resemble the free RelA spectrum (black). The RelA residues affected by the addition of the anthracyclines largely coincide with the DNA-binding region. Amino acid residues in RelA–DNA complex particularly affected by Epi were 50R, 56K, 62K, 78T, 79K, 88H, 89E, 119Q, 129A, and 157V in the DBD and 192A, 245H, 250I, and 268V in the DD (*Figure 5A*, left). Addition of Acla to RelA–DNA complexes caused precipitation of the protein, which made measurements difficult. Yet, residues 88H, 119Q, 129A, and 186N in the DBD and 245H in the DD could be analyzed and were altered (*Figure 5A*, right). Furthermore, addition of Epi to the RelA–DNA complex caused NMR chemical shift changes for some backbone amide signals in RelA (268V, 250I, and 50R) toward an unbound state (*Figure 5B*, left zoom-in), an effect that was not observed upon Acla addition (*Figure 5B*, right zoom-in). Because the final spectra are not identical (compare green spectra in *Figure 5B*), Epi and Acla are expected to have different effects on the RelA–DNA complex. Differences in binding likely reflect the distinct chemical moieties involved, particularly because Acla contains three sugar rings while Epi contains one. In fact, studies showed that the sugar moieties of the anthracyclines bind to the minor groove of the DNA (*Temperini et al., 2003*; *Chaires, 2015*; *Frederick et al., 1990*). This may be compatible with the 3D structure of RelA-κB-33 (PDB code 1RAM, *Figure 5A*, *Figure 4—figure supplement 3A*), with the TG sequences at the end of the DNA duplex and with the accessible minor groove being able to accommodate the anthracycline sugars and in close vicinity of DD. Indeed, our results suggest that Epi and Acla bind to RelA through their sugar parts (*Figure 5—figure supplement 1*). Notably, addition of Epi to RelA followed by addition of DNA led to different spectra (*Figure 5—figure supplement 2A*, compare red and green spectra), suggesting that the final complexes are slightly different. In the case of Acla spectra (*Figure 5—figure supplement 2B*, compare red and green spectra), the interpretation is compromised because addition of Acla to RelA leads to protein precipitation, and the final spectrum obtained after DNA addition (green) is therefore much weaker than its counterpart (red).

To further explore how anthracyclines affect the RelA–DNA complex we performed ITC competition experiments. The binding affinities of DNA to a pre-formed complex of RelA with Epi (980 nM)

**Table 1.** Summary of the data obtained by isothermal titration calorimetry (ITC).
The ligand (or titrant) was titrated to the analyte (or titrand) in the cell, for example Epi to DNA. The measured thermodynamic properties were: $K_D$, dissociation constant; $N$, stoichiometry of the titrant; $\Delta H$, enthalpy change; and $\Delta S$, entropy change. $T$ is the measurement temperature.

| | KD (µM) | *N* | $\Delta H$ (kJ/mol) | $-T\Delta S$ (kJ/mol) |
|---|---|---|---|---|
| Epi to DNA | 11.3 ± 4.6 | 1.83 ± 0.14 | −14.1 ± 2.2 | −14.2 |
| Acla to DNA | 11.7 ± 2.0 | 2.13 ± 0.07 | −38.4 ± 2.0 | 10.3 |
| DNA to RelA | 0.23 ± 0.03 | 0.90 ± 0.01 | 40.7 ± 0.9 | −78.6 |
| (DNA + Epi) to RelA | 1.9 ± 0.17 | 0.88 ± 0.04 | 59.4 ± 3.0 | −92.1 |
| DNA to (RelA + Epi) | 0.98 ± 0.12 | 0.81 ± 0.01 | −305 ± 9 | 271 |
| DNA to (RelA + Acla) | 0.74 ± 0.32 | 0.91 ± 0.07 | −335 ± 40 | 300 |
| Epi to RelA | ND. Too weak. | | | |
| Acla to RelA | ND. Too weak. | | | |

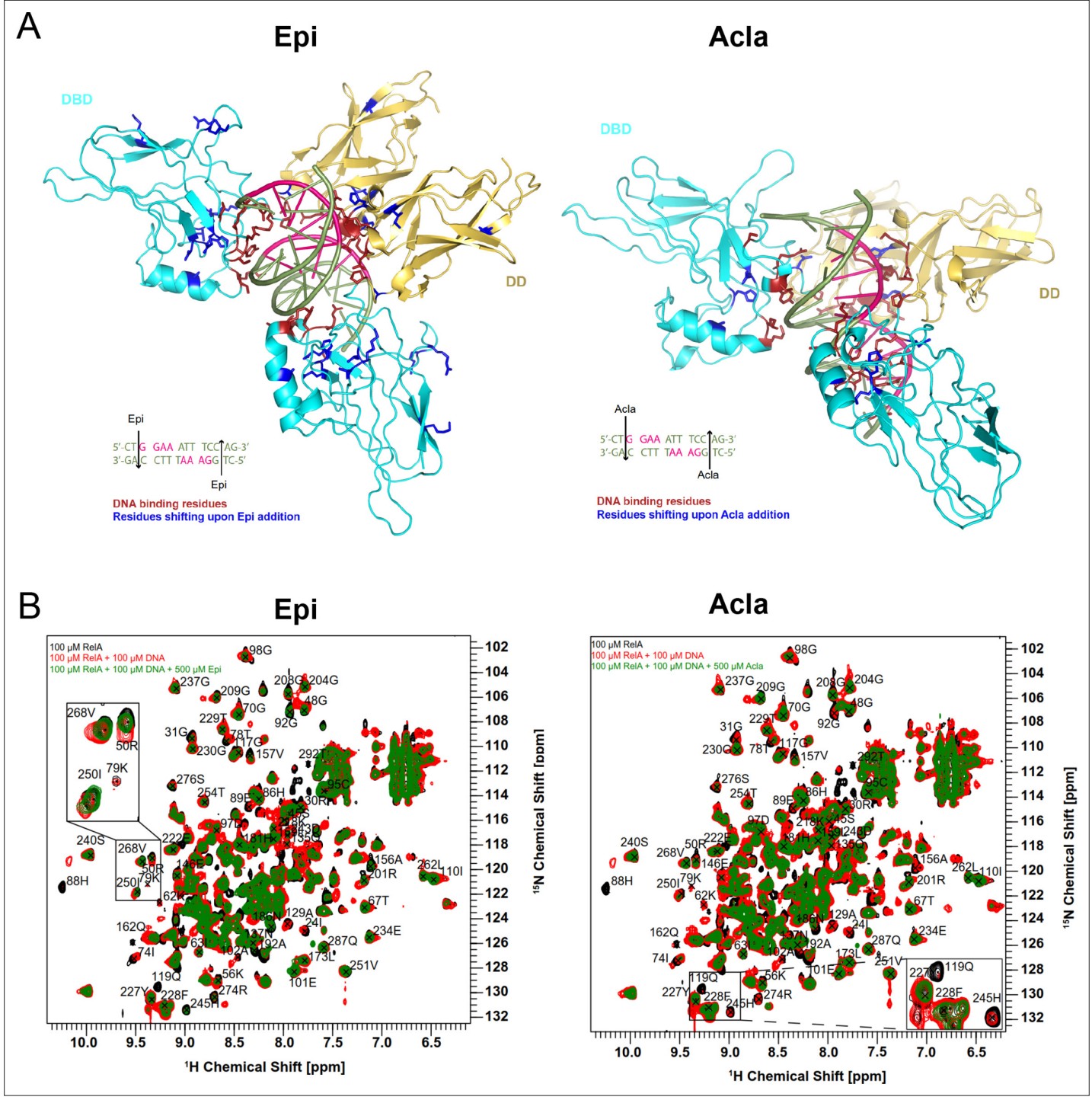

**Figure 5.** The RelA REL-homology domains (RHDs) of Epi and Acla are shown in a complex with the κ B-33 promoter. Epirubicin (Epi) and Aclarubicin (Acla) are capable of disturbing RelA–DNA binding as assessed by NMR chemical shift changes. (**A**) Crystal structure of RelA RHD, containing the DNA-binding domain (DBD) and the dimerization domain (DD), in complex with κ B-33 promoter DNA target (PDB code 1RAM). The RelA DBD and DD are represented as cartoon in cyan and yellow, respectively, whereas the DNA in olive green and pink. DNA highlighted in pink corresponds to the two half sites recognized by each RelA monomer. DNA-binding residues are depicted in red, whereas the amino acid residues that shift after addition of Epi (**A**, left) and Acla (**A**, right) are depicted in blue. (**B**) Superposition of 2D $^1$H,$^{15}$N NMR correlation spectra of 100 µM $^2$H, $^{15}$N-RelA$_{19-291}$-free dimer (black), and in the presence of 100 µM κ B-33 14-mer DNA duplex (red) and upon additional presence of 500 µM Epi (green, left) or 500 µM Acla (green, right). All spectra were recorded at 800 MHz, 20°C, 16 scans.

The online version of this article includes the following figure supplement(s) for figure 5:

*Figure 5 continued on next page*

*Figure 5 continued*

**Figure supplement 1.** Saturation transfer difference (STD)-NMR experiments to test the interaction of RelA with the anthracyclines Epirubicin (Epi), Aclarubicin (Acla), and Doxorubicinone (Doxo-none).

**Figure supplement 2.** Comparison of the effect of the order of addition in NMR-binding studies of Epirubicin (Epi) and Aclarubicin (Acla) with RelA and κB-33 DNA.

**Figure supplement 3.** NMR-binding studies on RelA$_{19-291}$ with Epirubicin (Epi) and Aclarubicin (Acla).

**Figure supplement 4.** Model for the regulation of nuclear factor kappa B (NF-κB) binding to target promoters by Epirubicin (Epi).

**Figure supplement 5.** Model of the effect of anthracyclines on RelA RHD binding to the κB-33 promoter DNA.

or Acla (740 nM) were weaker than the binding affinities of the DNA to the protein alone (230 nM) (*Table 1*). Moreover, the thermodynamic features of the interaction were different: binding of DNA to RelA is entropy driven and endothermic (positive Δ$H$), while binding of DNA to RelA incubated with anthracyclines is enthalpy driven and exothermic (negative Δ$H$). Surprisingly, when DNA with a twofold excess of Epi was titrated to the protein, the binding affinity decreased dramatically from a $K_D$ of 230 to 1900 nM but the reaction is similarly endothermic (*Table 1*). This suggests that when the anthracyclines–DNA complex is formed, binding to RelA is mainly affected.

We also tested a potential direct binding between anthracyclines and RelA. We did not observe any binding by ITC, likely because this method is not suitable for very weak interactions (*Table 1*). We did not observe significant RelA backbone amide changes after the addition of anthracyclines in 2D-TROSY NMR protein spectra, indicating that Epi and Acla bind very weakly to the protein (*Figure 5—figure supplement 3*). Binding to RelA, though weak, is mediated through the sugar part of the anthracyclines, as observed in saturation transfer difference (STD) experiments (*Figure 5— figure supplement 1A, B*). In agreement, Doxo-none, which lacks the sugar moiety, did not bind to RelA (*Figure 5—figure supplement 1C*).

## Discussion

We have studied the effect of anthracyclines on NF-κB controlled inflammatory responses. At very low concentrations, this anticancer family of drugs suppressed the production of pro-inflammatory signals in mouse macrophages. Using biophysical methods and NMR we found that anthracyclines directly inhibit the binding of the NF-κB family member RelA to the DNA in a manner that contributes to the resolution of inflammation. Taking advantage of a series of dedicated anthracycline variants with defined activities, we dissociated the DNA damaging from the histone eviction activities, therefore uncoupling the anti-inflammatory effect from the classical DDR.

Despite the general notion that anthracyclines induce NF-κB-dependent anti-apoptotic gene expression when used as chemotherapeutic drugs (*Wang et al., 1996*; *Arlt et al., 2001*; *Janssens et al., 2005*; *Van Antwerp et al., 1996*; *Wu et al., 1996*), Doxo and Dauno have also been described as repressors of NF-κB activity (*Campbell et al., 2004*; *Ho et al., 2005*). The outcomes of anthracycline treatments are likely dependent on cell- and tumor type. Using transcriptomics in macrophages, we found that only a very small proportion of inflammation-driven genes were upregulated by either Epi or Acla while none of the anti-apoptotic factors tested, including cellular inhibitor of apoptosis proteins (cIAPs), were affected. Instead, we detected downregulation of a broad range of NF-κB-induced genes in Epi- or Acla-treated macrophages. Even if the affinity of the binding to κB sites is not directly linked to transcriptional activity (*Mulero et al., 2017*) our results showed that Epi- and Acla treatments reduced the binding of RelA and of PolII to κB sites upon pro-inflammatory stimulation, which implies a novel role for the anthracyclines in counteracting NF-κB activity.

Because Epi is protective in sepsis through ATM-mediated signaling (*Figueiredo et al., 2013*), and due to the genotoxicity of this class of drugs, we asked whether cytokine regulation by anthracyclines depended on the induction of DDRs. We observed that pro-inflammatory gene expression orchestrated by NF-κB was still downregulated by Acla, Doxo-none, and the newly developed diMe-Doxo, a group of anthracyclines that do not cause DNA damage in the concentrations used. Furthermore, cytokine suppression by anthracyclines was observed when we used an ATM inhibitor and *Atm$^{-/-}$* BMDMs, indicating independence from the DDR. Finally, other DNA damaging drugs such as Eto did not modulate cytokine production. Together, these observations suggested that regulation of

NF-κB-dependent transcription by anthracyclines does not rely on the induction of DNA damage. Whereas anthracyclines are believed to work in cancer because of the induction of DNA double-strand breaks, anthracyclines that lack DNA damaging activity, such as Acla, are also effective in the treatment of acute myeloid leukemia (*Qiao et al., 2020*). Clearly, other activities, such as chromatin damage caused by histone eviction, contribute to the anti-tumor role of the anthracyclines (*Pang et al., 2013*; *Yang et al., 2013*). It is likely that regulation of NF-κB also proves to be important for the therapeutic efficacy of these drugs.

But can we understand how anthracyclines control NF-κB? Both Epi and Acla interacted with DNA bases within the κB motif, probably intercalating between the TG bases, which disturbed the complexes between RelA and its DNA-binding sites (models of these interactions are depicted in *Figure 5—figure supplement 4* and *Figure 5—figure supplement 5*). The anthracyclines sugars also interact weakly with RelA, promoting the formation of a ternary complex between anthracyclines, DNA and RelA. The RelA residues targeted by Epi and Acla localized within the RHD, which corroborated the reporter assays. Whereas Epi and Acla perturbed the binding of RelA to DNA differently, the structural results did not explain why Acla is the strongest regulator of NF-κB-dependent transcription. The extra sugars in Acla may be responsible for the stronger interactions but it will be important to do a careful dissection of the structural features contributing to the efficient Acla-mediated NF-κB transcriptional suppression. Not surprisingly, Doxo-none – that lacks the sugar moiety – did not bind to RelA and was a weaker modulator of *TNF* transcription when compared with Epi, Doxo, and Acla. However, diMe-Doxo, with a sugar moiety that resembles both Epi (contains only one sugar) and Acla (the amine group is dimethylated), was also weaker than the classical anthracyclines at regulating the genes tested. We are currently investigating its RelA and DNA-binding properties to uncover potential new rules affecting complex formation. In cells, the engagement of RelA and other NF-κB factors with κB sites is highly dynamic and regulated by chromatin organization and other nuclear factors (*Mulero et al., 2017*). It will also be important to explore promoter-specific effects.

To conclude, we showed that the control of inflammatory responses by anthracyclines does not require their DNA damage effects. The novel role in NF-κB suppression, although with different efficacies, is common to all anthracyclines tested independently of DNA damage capabilities. We thus uncovered a new mechanism of action for the anthracycline class of anti-cancer drugs that can inform applications in situations of uncontrolled inflammatory responses, such as sepsis. Limiting pro-inflammatory gene expression through NF-κB modulation may have been an overlooked aspect of Epi, Doxo, and Acla with a fundamental role in the clinical success of these drugs. Future therapeutic strategies are expected to focus on anthracyclines without DNA damaging activities, in an attempt to overcome the severe side effects of these drugs. Indeed, diMe-Doxo and Acla, both lacking DNA damage activity, are not cardiotoxic and do not induce second tumors (*Qiao et al., 2020*). These could therefore be more suitable for controlling sepsis and other inflammatory responses. Importantly, low doses seem to be sufficient to prompt this regulatory effect.

## Methods

### Key resources table

| Reagent type (species) or resource | Designation | Source or reference | Identifiers | Additional information |
|---|---|---|---|---|
| Strain, strain background (male *Mus musculus*) | C57BL/6J | Instituto Gulbenkian de Ciência | JAX:000664 | |
| Strain, strain background (male *Mus musculus*, background C57BL/6J) | *Atm⁻/⁻* | *Borghesani et al., 2000* doi:10.1073/pnas.97.7.3336 | | From Frederick W. Alt (Harvard Medical School, Boston, MA, USA) |
| Cell line (*Homo sapiens*) | HEK293 | ATCC | CRL-1573 | |
| Cell line (*Homo sapiens*) | THP-1 | TCC | TIB-202 | |
| Antibody | Anti-RelA (rabbit polyclonal) | Santa Cruz | sc-372 | 1:2000 |
| Antibody | Anti-RNAPolII (mouse monoclonal) | Santa Cruz | sc-17798 | 1:1000 |

*Continued on next page*

*Continued*

| Reagent type (species) or resource | Designation | Source or reference | Identifiers | Additional information |
|---|---|---|---|---|
| Antibody | Anti-IκBα (rabbit polyclonal) | Santa Cruz | sc-371 | 1:2000 |
| Antibody | Anti-β-actin (mouse monoclonal) | Santa Cruz | sc-58673 | 1:4000 |
| Peptide, recombinant protein | Lipopolysaccharide (LPS) from *E. coli* 055:B5 | Santa Cruz | sc-221855B | |
| Peptide, recombinant protein | Human TNF-alpha | Peprotech | 300-01A | |
| Commercial assay or kit | AlamarBlue Cell Viability Reagent | Invitrogen | DAL1100 | |
| Commercial assay or kit | RNeasy Mini Kit | Qiagen | 50974106 | |
| Commercial assay or kit | Taq Universal SYBR Green Supermix | Bio-Rad | 1725125 | |
| Commercial assay or kit | Superscript II | Invitrogen | 18064014 | |
| Commercial assay or kit | Lipofectamine 3000 Transfection Reagent | Invitrogen | L3000-001 | |
| Commercial assay or kit | Luciferase Assay System | Promega | PROME1500 | |
| Commercial assay or kit | Mouse TNF-α ELISA MAX Standard | BioLegend | 430902 | |
| Commercial assay or kit | Mouse IL-6 ELISA MAX Standard | BioLegend | 431302 | |
| Commercial assay or kit | Mouse IL-12(p40) ELISA MAX Standard | BioLegend | 431602 | |
| Commercial assay or kit | LEGEND MAX Mouse IFN-β ELISA Kit | BioLegend | 439407 | |
| Commercial assay or kit | Mouse CXCL10 DuoSet ELISA | R&D Systems | DY466 | |
| Commercial assay or kit | Human/Mouse/Rat Phospho-Histone H2AX | R&D Systems | KCB2288 | |
| Commercial assay or kit | CometAssay Kit 25 × 2 well slides | Trevigen | 4250-050K | |
| Chemical compound, drug | Epirubicin | Target Mol | 282T0125 | |
| Chemical compound, drug | Aclarubicin | FOCUS Biomolecules | 21910-1099 | |
| Chemical compound, drug | Doxorubicin | FOCUS Biomolecules | 21910-2130 | |
| Chemical compound, drug | Etoposide | Sigma-Aldrich | E1383 | |
| Chemical compound, drug | DMSO | Sigma-Aldrich | D2650 | |
| Chemical compound, drug | KU-55933 | Sigma-Aldrich | SML1109 | |
| Chemical compound, drug | MG-132 | Sigma-Aldrich | M7449 | |
| Chemical compound, drug | Actinomycin D | Sigma-Aldrich | A9415 | |
| Chemical compound, drug | D-(+)-Glucose solution | Sigma-Aldrich | G8769 | |
| Chemical compound, drug | cOmplete, EDTA-free | Roche | 11873580001 | |
| Chemical compound, drug | DL-Dithiothreitol-d10 | Cortecnet | CD4035P1 | |
| Chemical compound, drug | Dimethylsulfoxide D6 | Eurisotop | D010 | |
| Chemical compound, drug | Tris-D11 | Cortecnet | CD4035P1 | |
| Other | Ni-NTA Agarose | Qiagen | 30210 | Purification resin |
| Other | HiLoad 16/600 Superdex | Merck | GE28-9893-33 | Purification resin |
| Other | Superdex 200 10/300 GL | Merck | GE17-5175-01 | Purification resin |

*Continued on next page*

*Continued*

| Reagent type (species) or resource | Designation | Source or reference | Identifiers | Additional information |
|---|---|---|---|---|
| Other | RPMI 1640 Medium, no glutamine | Life Technologies | 31870-025 | Component of cell culture media |
| Other | DMEM, high glucose, pyruvate | Life Technologies | 41966-052 | Component of cell culture media |
| Other | Fetal bovine serum (FBS) | Life Technologies | 10500-064 | Component of cell culture media |
| Other | Penicillin–streptomycin | Life Technologies | 15140-122 | Component of cell culture media |
| Other | Sodium pyruvate | Life Technologies | 11360-039 | Component of cell culture media |
| Other | L-Glutamine | Life Technologies | 25030-024 | Component of cell culture media |
| Other | Non-essential aminoacids solution | Life Technologies | 11140-035 | Component of cell culture media |
| Other | HEPES buffer | Life Technologies | 15630-056 | Component of cell culture media |
| Other | 2-Mercaptoethanol | Life Technologies | 31350-010 | Component of cell culture media |

## Resources availability

Information and requests for resources and reagents should be directed to and will be fulfilled by the Lead Contacts.

## Mice

All animal studies were performed in accordance with Portuguese regulations and approved by the Instituto Gulbenkian de Ciência ethics committee and DGAV. $Atm^{-/-}$ (**Borghesani et al., 2000**) and C57BL/6J control mice (JAX:000664, bred at IGC) were bred and maintained under specific pathogen-free conditions at the Instituto Gulbenkian de Ciência with 12 hr light/12 hr dark cycle, humidity 50–60%, ambient temperature 22 ± 2°C and food and water ad libitum. Male mice of 8–12 weeks old were used in the experiments and, following sample size calculation, age-matched mice were randomly assigned to experimental groups.

## Primary cell cultures

For the generation of bone marrow-derived macrophages, total bone marrow cells were flushed from femurs and tibiae, counted and seeded ($3 \times 10^6$ cells/ml) in RPMI 1640 supplemented with 10% (vol/vol) fetal bovine serum (FBS), 0.2% (vol/vol) penicillin–streptomycin, 1% (vol/vol) sodium pyruvate, 1% (vol/vol) L-glutamine, 1% (vol/vol) non-essential aminoacids, 1% (vol/vol) HEPES (N-2-hydroxyethylpiperazine-N-2-ethane sulfonic acid) buffer and 0.05 mM of 2-Mercaptoethanol (all Gibco, Life Technologies) supplemented with 30% conditioned medium from mouse macrophage colony-stimulating factor-producing L929 cells. On day 7, adherent BMDM cells were collected, counted, reseeded in 96-well plates ($5 \times 10^4$/well) and treated and/or challenged as indicated. HEK293 cells were obtained from ATCC and cultured in DMEM (Dulbecco's Modified Eagle Medium) supplemented with 10% (vol/vol) FBS and 1% (vol/vol) penicillin–streptomycin (all Gibco, Life Technologies). Cells were cultured in a humidified atmosphere containing 5% $CO_2$.

## Main reagents

Epirubicin, Aclarubicin, Doxorubicin, Daunorubicin, Doxorubicinone, and Dimethyl-doxorubicin were dissolved in phosphate-buffered saline (PBS) at 1 mg/ml, stored at −80°C and used at the indicated final concentrations. Anthracyclines were from various commercial sources except for Dimethyl-doxorubicin, a gift from Jacques Nefjees. Etoposide (Sigma-Aldrich) was dissolved in DMSO and used at indicated final concentrations. MG-132 (Sigma-Aldrich) was dissolved in DMSO and used at a final

concentration of 10 µM. ATM inhibitor KU-55933 (Sigma-Aldrich) was dissolved in DMSO and used at a final concentration of 5 µM. Human recombinant TNF (PeproTech) was dissolved in RPMI and used at a final concentration of 10 ng/ml for the indicated times. Actinomycin D (ActD, Sigma-Aldrich) was dissolved in DMSO and used at a final concentration of 5 µg/ml for the indicated times. LPS (from *E. coli* 055:B5, Santa Cruz) was dissolved in RPMI and used at a final concentration of 100 ng/ml. PFA (Paraformaldehyde)-fixed *E. coli* were prepared as before (*Moura-Alves et al., 2011*) in house and added to the medium at a ratio of 20 bacteria per cell. AlamarBlue cell viability assay (Invitrogen) was used to determine cell viability according to the manufacturer's instructions.

### *E. coli*-induced sepsis model

This model has been described in detail before (*Colaço et al., 2021*). Briefly, a starter culture from a single *E. coli* colony was grown overnight at 37°C with agitation in Luria-Bertani broth (LB). The following morning, the culture was diluted 1:50 in LB and incubated for 2.5 hr until late exponential phase was reached ($OD_{600\ nm}$ = 0.8–1.0). The culture was then centrifuged at 4400 × *g* for 5 min at room temperature, washed and resuspended in PBS to $OD_{600\ nm}$ = 4.5–5.0, corresponding to 1–2 × $10^9$ CFU/ml. This bacterial suspension was immediately injected intraperitoneally (200 µl/mouse) in mice using a 27G-needle, always in the morning. Epirubicin or Aclarubicin were dissolved in PBS and injected intraperitoneally (200 µl/mouse) at 0.5 and 0.73 µg/g body weight, respectively, at the time of infection. Mice were sacrificed at 8 hr by $CO_2$ inhalation and blood was collected by cardiac puncture. Serum was collected after centrifuging the blood at 1600 × *g* for 5 min.

## Cytokine production measurement

Mouse sera and cell culture supernatants were collected at indicated time points and TNF, Cxcl10, IL6, IL-12p40, and IFNβ production quantified using ELISA kits (BioLegend and R&D Systems) according to the manufacturer's instructions. qRT-PCR.

Total RNA was isolated from BMDM using the RNeasy Mini Kit (Qiagen) and reverse transcribed with Superscript II reverse transcriptase (Invitrogen) using oligo(dT)12–18 primers. Specific RNA specimens were quantified by PCR reaction using SYBRgreen Supermix (Bio-Rad) on the QuantStudio7 Flex real-time PCR system (Applied Biosystems). Cycling parameters were as follows: 95°C for 10 min, followed by 40 cycles of PCR reactions at 95°C for 30 s and 60°C for 1 min. The relative expression levels of the genes assayed were calculated using the comparative cycle threshold Ct (ΔΔCt) method. Initially, the Ct value of each gene was normalized to the corresponding Ct value of Gapdh for the same sample to obtain the relative threshold cycle (ΔCt). The ΔCts were then transformed into relative expression by calculating $2^{-\Delta Ct}$. Each sample was then normalized to the average of the control sample for the same gene (ΔΔCt). The ΔΔCt of the samples assayed was expressed as relative fold change. Primer sequences can be found in Appendix 1.

## RNA-seq and data analysis

Total RNA was extracted as described above and quality was assessed using the AATI Fragment Analyzer. Samples with RNA Quality Number (RQN) >8 and clearly defined 28S and 18S peaks were further used for preparation of mRNA libraries, which were pooled and sequenced (75 bp, single end) using NextSeq500. The quality of the sequences was assessed using FASTQC and MultiQC before the alignment (*Ewels et al., 2016*). Sequences were aligned against the *Mus musculus* genome version 89, with the annotation file for the genome version 89, both from Ensembl. The alignment was done with STAR (*Dobin et al., 2013*), using the default parameters and including the GeneCounts option. The files from GeneCounts were imported to R (version 3.5.1), taking into account the strandness associated with the sequencing protocol. DESeq2 (version 1.22.1) (*Love et al., 2014*) was used for the downstream analysis. Heatmaps were created with data normalized from raw counts through Regularized Log Transformation (rlog) (*Love et al., 2014*). The log2FC was shrunk using the 'ashr' (Adaptive SHrinkage) package (*Stephens, 2017*) and genes were considered differentially expressed when the p value was below 0.05 after adjusting using false discovery rate. Gene Information was obtained from org.Mm.eg.db. Functional clustering was performed using the DAVID Gene Functional Classification Tool (https://david.ncifcrf.gov). The RNA-seq datasets and scripts generated during this study are available on GitHub: https://github.com/andrebolerbarros/Chora_etal_2022/, (copy archived at swh:1:rev:d86a4302612c313875b919e3897bec15310b6895; *Barros, 2022*).

## KO studies

We performed CRISPR/Cas9-mediated KOs of the NF-κB subunit RelA in HEK293 cells (ATCC catalog number CRL-1573; cells were regularly tested for mycoplasma with a PCR-based detection assay). We used commercially available double nickase plasmids from Santa Cruz: RelA nickase plamid sc-400004-NIC-2; and scrambled control RNA sequence sc-437281. The double nickase plasmids consist of a pair of plasmids each encoding a D10A mutated Cas9 nuclease and a target-specific 20 nt guide RNA (gRNA). Plasmid transfection was performed as described below using the amounts of DNA specified by Santa Cruz. The resulting population was tested (at week 1 of KO) for RelA protein expression and RNA levels. The primary antibody used for WB detection was anti RelA antibody sc-372 (Santa Cruz) at 1:2000 dilution (the same antibody used in immunofluorescence below). The primers used in qRT-PCR were: for *RELA*, forward primer ATGTGGAGATCATTGAGCAGC and reverse primer CCTGGTCCTGTGTAGCCATT; for *GAPDH*, forward primer GAGTCAACGGATTTGGTCGT and reverse primer TTGATTTTGGAGGGATCTCG; and for *TNF*, forward primer CCGAGTGACAAGCCTG TAGC and reverse primer GAGGACCTGGGAGTAGATGAG.

## Antibodies used in western blot and immunofluorescence

The following antibodies were used for the specific detection of: IκBα, polyclonal sc-371, Santa Cruz, used at dilution 1:500 in WB; RelA, polyclonal sc-372, Santa Cruz, used at dilution 1:100 (IF); and β-actin monoclonal (2Q1055) sc-58673, Santa Cruz, used at dilution 1:500 (WB). Primary antibodies were detected using HRP-conjugated secondary antibodies (Cell Signaling).

## Transient transfection and reporter assay

N-terminal Myc-tagged RelA, RelA (2-320)/VP16, TET/RelA (268-551), and TET/VP16 expression vectors, the NF-κB firefly luciferase reporter construct, and the tetracycline operon (tet°) firefly luciferase reporter (tet°-luc), were previously described (*Anrather et al., 1999*). The κB sequences in the κB-luciferase reporter are: 5'-TGCTGGGAAACTTTC-3' and 5'-TGCTGGGAATTCCTC-3', closely matching the originally described consensus motif GGGRNYYYCC (in which R is a purine, Y is a pyrimidine, and N is any nucleotide, *Chen et al., 1998*). The promoter sequences in this vector are from a porcine adhesion molecule (E-selectin or ELAM1, expressed on the surface of activated endothelial cells). In addition to the two RelA consensus-like sequences, there is a third sequence believed to work as an enhancer. The pSV-β-galactosidase reporter consists of the *lacZ* gene from *E. coli* under the control of the SV40 early promoter and enhancer. Transient transfections using HEK293 cells (same cells as described above) were performed with the lipofectamine 3000 transfection reagent (Invitrogen). Twenty-four hours after transfection, cells were either pre-exposed to Epirubicin for 1 hr prior to TNF stimulation or treated with Epirubicin for 16 hr in the RelA overexpression assays. After incubation, cells were lysed and firefly luciferase and β-galactosidase activity were measured using the Luciferase Assay System (Promega) and Galacto-Ligh System (Invitrogen), respectively, following the manufacturer's instructions.

## Chromatin immunoprecipitation

ChIP was performed on HEK293 cells as previously described (*de Almeida et al., 2011*). Antibodies against RelA (sc-372, Santa Cruz) and RNAPolII (sc-899, Santa Cruz) were used for immunoprecipitation.

## Comet assay

Comet assay was performed in THP-1 cells (ATCC catalog number TIB-202) using the CometAssay Kit 25 × 2 well slides (Trevigen catalog number 4250-050K). A step-by-step protocol detailing the exact procedures and all the materials used is available online from Protocol Exchange (Alkaline Comet Assay using the monocytic cell line THP-1, https://doi.org/10.21203/rs.2.11936/v2).

## phosphoH2AX quantification

Phosphorylation of histone H2AX at serine 139 was quantified by cell-based ELISA using the kit Human/Mouse/Rat Phospho-Histone H2AX (R&D catalog number KCB2288) according to the manufacturer's instructions.

## Quantification and statistical analysis

Data are expressed as mean values ± standard deviation. Mann–Whitney test was used for pairwise comparisons and two-way analysis of variance with Tukey test was used for multiple comparisons. Statistical analysis was performed with GraphPad Prism 6.0 (GraphPad Software). The number of subjects used in each experiment is defined in figure legends. The following symbols were used in figures to indicate statistical significance: $*p < 0.05$; $**p < 0.01$; $***p < 0.001$; $****p < 0.0001$.

## Purification of recombinant RHD RelA

A human RelA construct (residues 19–291) was prepared by subcloning into pETM11 vector, which contains an N-terminal His6-tag and a TEV protease cleavage site. The new construct contains the N-terminal DBD (residues 19–191) and the C-terminal DD (residues 192–291). The plasmids were transformed into *E. coli* strain BL21 (DE3) cells and cultured overnight at 20°C in LB media supplemented with 100 μg/ml kanamycin.

For the preparation of uniformly labeled $^2H$ (~100%) RHD RelA, $^{15}N$ (99%)-labeled protein was expressed at 37°C using M9 minimal medium containing $^{15}NH_4Cl$, [$^{12}C$]D-d7-glucose(2 g/l) (97% *D*, Sigma-Aldrich) in 100% $D_2O$. A standard protocol of sequential precultures for better $D_2O$ adaptation over a 3-day period was followed to increase the yield of protein expression in 100% $D_2O$. On the first day, a 25-ml preculture in LB medium was prepared and grown overnight at 37°C. The following day, a preculture of 50 ml M9 minimal medium in $H_2O$ was inoculated with 1 ml of the overnight LB preculture and grown at 37°C. After some hours, when the preculture reached an optical density at 600 nm ($OD_{600}$) close to 0.6, it was spun down for 10 min at $3202 \times g$. The cells were resuspended in 1 ml of M9 medium in 100% $D_2O$ and used for the inoculation of 100 ml of M9 medium in 100% $D_2O$, such that the $OD_{600}$ was 0.1–0.15. This small culture was left overnight at 37°C. The next day, this culture was added to 900 ml of M9 medium in 100% $D_2O$. All cultures in minimal media were induced at $OD_{600}$ of 0.8 with 0.5 mM of IPTG overnight at 20°C.

After overnight induction, cell pellets were lysed by sonication in lysis buffer (50 mM) Tris–HCl pH 8, 300 mM NaCl, 5 mM imidazole, 5 mM mercaptoethanol, 0.025 mg/ml DNAse I, 0.1 mg/ml lysozyme, 2.5 mM $MgSO_4$, 0.1% NP-40, and 1 pill of protease inhibitor EDTA-free (cOmplete Tablets, Mini EDTA free, Roche) per 30 ml lysate. The cell lysate was centrifuged at $60,000 \times g$ for 30 min at 4°C. After filtration, the His-tagged protein in the supernatant was loaded on an IMAC (Immobilized Metal Affinity Chromatography). The supernatant was applied to Ni-NTA resin (Qiagen) previously equilibrated with 3 column volumes of buffer A (50 mM Tris–HCl pH 8, 300 mM NaCl, 5 mM imidazole, 5 mM mercaptoethanol). Bound protein was washed with 3 column volumes of buffer A and unspecific bound protein was washed away with 3 column volumes of Wash Buffer (50 mM Tris–HCl pH 8, 1 M NaCl, 5 mM imidazole, and 5 mM mercaptoethanol). His$_6$-tagged protein was eluted using elution buffer (50 mM Tris–HCl pH 8, 300 mM NaCl, 300 mM imidazole, and 5 mM mercaptoethanol). The affinity His-tag was removed from the protein by TEV (1:5 protein:TEV ratio) cleavage during dialysis into 50 mM Tris–HCl pH 8, 300 mM NaCl, and 5 mM mercaptoethanol buffer overnight at 4°C. The cleaved tag and TEV protease were removed from the target protein using a second IMAC step in dialysis buffer. The fractions containing RelA were pooled, concentrated and further purified by size-exclusion chromatography (SEC) using a Superdex 75 Hiload 16/60 column (S75, GE Healthcare, Merck). The SEC buffer used was 50 mM Tris–HCl pH 7.5, 150 mM NaCl, 1 mM EDTA, and 5 mM Dithiothreitol (DTT). For the $^2H$, $^{15}N$-labeled RelA, prior to the SEC purification step, 2 M urea were added to the protein sample for 1 hr, in order to enhance the proton chemical exchange. The final yields were 12.5 mg for $^2H$, $^{15}N$ RelA and 52 mg for unlabeled RelA per liter of cell culture.

For NMR experiments, all protein samples were exchanged by successive concentration/dilution steps into NMR buffer (100 mM d11-Tris–HCl [Cortecnet] pH 7.5, 75 mM NaCl, and 5 mM d10-DTT [Cortecnet], 90% $H_2O$/10% $D_2O$). The protein concentrations were calculated using the absorption at 280 nm wavelength by using molar extinction coefficients of 17420 $M^{-1}$ $cm^{-1}$ for $RelA_{19-291}$.

## NMR spectroscopy

One-dimensional (1D) $^1H$ NMR experiments were recorded using a WATERGATE pulse sequence at 25°C on a Bruker AvanceIII 800 MHz spectrometer equipped with a cryogenic TCI-probehead ($^1H$, $^{31}P$, $^{13}C$, and $^{15}N$) with Z-gradients. 1D $^1H$ experiments were performed using a WATERGATE pulse sequence with 32k time domain and 128 scans in 100 mM d11-Tris–HCl pH 7.5, 75 mM NaCl, 5 mM

d10-DTT, and 10% $D_2O$. STD experiments were recorded using an interleaved pulse program with on-resonance protein irradiation at 0.15 ppm for Epirubicin, 0.5 ppm for Aclarubicin, and 0.6 ppm for Doxorubicinone and off-resonance irradiation at –5 ppm with 2 s effective irradiation, using 800 scans and 32k time domain points (600 MHz). Each experiment was performed using 500 µM of compound 10 µM of unlabeled protein. Reference STD experiments without protein were performed at the same conditions, using the same irradiation regions. Spectra were processed using TOPSPIN 3.2 (Bruker Biospin, Rheinstetten, Germany).

NMR-binding studies were performed at 25°C using 100 µM $^2H$(~100%), $^{15}N$-labeled RelA$_{19-291}$ dimer in a 100 mM d11-Tris–HCl buffer (pH 7.5, 75 mM NaCl, 5 mM d10-DTT, and 10% $D_2O$) by adding compound to a final concentration of 500 and/or 100 µM of duplex-DNA, and monitoring the changes by $^1H$, $^{15}N$ TROSY experiments. A reference experiment was performed under the same conditions with the same volume of DMSO-$d_6$ (Eurisotop) as used for the compound titration. Changes in the DNA structure were detected by recording NMR Imino NOESY spectra at 10°C, using 100 µM duplex-DNA, to which compound (Epi and Acla) to a final concentration of 500 and 50 µM $^2H$(~100%), $^{15}N$-labeled RelA$_{19-291}$ dimer in a 100 mM d11-Tris–HCl buffer (pH 7.5, 75 mM NaCl, 5 mM d10-DTT, and 10% $D_2O$) were added in a different order.

Chemical shift assignment of RelA$_{19-291}$ was obtained at 950 MHz and 25°C using TROSY versions of 3D HNCACB, HNCA, HN(CO)CA, HN(CA)CO, and HNCO experiments (*Sattler et al., 1999*, *Zhang et al., 1994*) on a 200 µM $^2H$(~100%), $^{13}C$, $^{15}N$-labeled RelA$_{19-291}$ dimer in 100 mM d11-Tris–HCl buffer (pH 7.5, 75 mM NaCl, 5 mM d10-DTT, and 10% $D_2O$) based on the assignment of three similar constructs of a $^{15}N$-labeled, perdeuterated RelA RHR (residues 19–325) in complex with perdeuterated p50 RHR (residues 37–363), a RelA DBD (residues 19–191), and a RelA DD (residues 190–321) in complex with deuterated p50 DD (residues 245–350) (*Mukherjee et al., 2016*). Assignment of RelA bound to DNA was performed by following the resonances during the DNA titration and confirmed by using a TROSY version of 3D HNCA on a 200 µM $^2H$(~100%), $^{13}C$, $^{15}N$-labeled RelA$_{19-291}$ dimer with 100 µM duplex 14-mer DNA in 100 mM d11-Tris–HCl buffer (pH 7.5, 75 mM NaCl, 5 mM d10-DTT, and 10% $D_2O$) obtained at 950 MHz and 25°C. All datasets were processed using NMRPipe (*Delaglio et al., 1995*) and analyzed with CCPN analysis 2.4.2 (*Vranken et al., 2005*).

## Static light scattering

SLS experiments were performed at 30°C using a Viscotek TDA 305 triple array detector (Malvern Instruments) downstream to an Äkta Purifier (GE Healthcare) equipped with an analytical size-exclusion column (Superdex 75 or 200 10/300 GL, GE Healthcare, Merck) at 4°C. The samples were run in 50 mM Tris–HCl pH 7.5, 150 mM NaCl, 1 mM EDTA, and 5 mM DTT with a concentration of 2 mg/ml for the protein and at ratios of 1:2 protein:DNA and of 1:2:6 protein:DNA:anthracyclines at a flow rate of 0.5 ml/min. The molecular masses of the samples were calculated from the refractive index and right-angle light-scattering signals using Omnisec (Malvern Instruments). The SLS detector was calibrated with a 4 mg/ml bovine serum albumin (BSA) solution with 66.4 kDa for the BSA monomer and a d$n$/d$c$ value of 0.185 ml/g for all protein samples.

## Isothermal titration calorimetry

ITC measurements were carried out at 25°C using a MicroCal PEAQ-ITC (Malvern Instruments Ltd). The titrations were performed in 50 mM HEPES pH 8.0, 100 mM NaCl, and 1 mM mercaptoethanol and 1% DMSO. The calorimetric titration consisted of 19 injections of 1.5 µl of a 125 µM DNA sample, into the reaction cell containing 400 µl of 25 µM RelA or to 25 µM RelA with 200 µM Epirubicin/Aclarubicin, at a stirring speed of 1000 rpm. The heat of dilution was obtained by titrating DNA into the sample cell containing only buffer and this was subsequently subtracted from each experimental titration. For evaluating the effect of Epirubicin bound to DNA on binding to RelA, a calorimetric titration consisted of 19 injections of 1.5 µl of a 125 µM DNA with 250 µM Epirubicin mixture, into the reaction cell containing 400 µl of 25 µM RelA. For the determination of the binding affinity of the compounds to the DNA, a calorimetric titration was performed consisting of 19 injections of 1.5 µl of a 500 µM compound sample, into the reaction cell containing 400 µl of 50 µM DNA. The heat of dilution was obtained by titrating compound into the sample cell containing only buffer and this was subsequently subtracted from each experimental titration. For the determination of the binding affinity of the DNA to the protein, a calorimetric titration was performed consisting of 19 injections of 1.5 µl of a 125 µM

DNA sample, into the reaction cell containing 400 µl of 25 µM RelA. The ITC data were analyzed using the MICROCAL PEAQ-ITC analysis software provided by Malvern.

## Acknowledgements

We are grateful to the Genomics Unit, Bioinformatics, and the Animal House at IGC. We thank Margarida Gama-Carvalho for the critical revision of the manuscript. This work was supported by the European Commission Horizon 2020 (ERC-2014-CoG 647888-iPROTECTION) and by Fundação para a Ciência e Tecnologia (FCT: PTDC/BIM-MEC/4665/2014 and EXPL/MED-IMU/0620/2021). SW is funded by the Deutsche Forschungsgemeinschaft, DFG, project number WE 4971/6-1, the Excellence Cluster Balance of the Microverse (EXC 2051; 390713860), and the Federal Ministry of Education and Research (BMBF) project number 01EN2001.

## Additional information

### Funding

| Funder | Grant reference number | Author |
| --- | --- | --- |
| H2020 European Research Council | 647888 | Angelo Ferreira Chora |
| Fundação para a Ciência e a Tecnologia | PTDC/BIMMEC/ 4665/2014 | Angelo Ferreira Chora |
| Fundação para a Ciência e a Tecnologia | EXPL/MED-IMU/0620/2021 | Ana Neves-Costa |
| Fundação para a Ciência e a Tecnologia | EXPL/MED-OUT/0745/2021 | Ana Neves-Costa |

The funders had no role in study design, data collection, and interpretation, or the decision to submit the work for publication.

### Author contributions

Angelo Ferreira Chora, Conceptualization, Data curation, Formal analysis, Investigation, Methodology; Dora Pedroso, Eleni Kyriakou, Henrique Colaço, Data curation, Formal analysis, Validation, Investigation, Methodology; Nadja Pejanovic, Tiago Velho, Catarina F Moita, Data curation, Validation, Investigation, Methodology; Raffaella Gozzelino, Pedro Pereira, Silvia Carvalho, Filipa Batalha Martins, João A Ferreira, Sérgio Fernandes de Almeida, Investigation, Methodology; André Barros, Data curation, Formal analysis, Investigation, Methodology; Katharina Willmann, Isa Santos, Validation, Investigation, Methodology; Vladimir Benes, Data curation, Methodology; Josef Anrather, Miguel P Soares, Resources; Sebastian Weis, Visualization; Arie Geerlof, Resources, Methodology; Jacques Neefjes, Resources, Investigation, Methodology; Michael Sattler, Conceptualization, Data curation, Investigation, Methodology; Ana C Messias, Conceptualization, Resources, Supervision, Validation, Investigation, Visualization, Methodology, Writing – original draft, Writing – review and editing; Ana Neves-Costa, Conceptualization, Data curation, Formal analysis, Supervision, Validation, Investigation, Visualization, Methodology, Writing – original draft, Project administration, Writing – review and editing; Luis Ferreira Moita, Conceptualization, Data curation, Formal analysis, Supervision, Funding acquisition, Validation, Investigation, Methodology, Writing – original draft, Project administration, Writing – review and editing

### Author ORCIDs

Dora Pedroso http://orcid.org/0000-0002-5735-5094
Henrique Colaço http://orcid.org/0000-0002-2026-0163
Tiago Velho http://orcid.org/0000-0002-0455-8189
Catarina F Moita http://orcid.org/0000-0002-9910-2343
Sérgio Fernandes de Almeida http://orcid.org/0000-0002-7774-1355
Vladimir Benes http://orcid.org/0000-0002-0352-2547
Sebastian Weis http://orcid.org/0000-0003-3201-2375

Miguel P Soares http://orcid.org/0000-0002-9314-4833
Jacques Neefjes http://orcid.org/0000-0001-6763-2211
Michael Sattler http://orcid.org/0000-0002-1594-0527
Ana C Messias http://orcid.org/0000-0002-5449-9922
Ana Neves-Costa http://orcid.org/0000-0001-6506-7829
Luis Ferreira Moita http://orcid.org/0000-0003-0707-315X

## Ethics

All animal studies were performed in accordance with Portuguese regulations and approved by the Instituto Gulbenkian de Ciência ethics committee and DGAV (A011_2019).

## Decision letter and Author response

Decision letter https://doi.org/10.7554/eLife.77443.sa1
Author response https://doi.org/10.7554/eLife.77443.sa2

## Additional files

### Supplementary files
• MDAR checklist

### Data availability

RNA-seq raw data of Murine Bone Marrow Derived Macrophages (BMDM's) stimulated with LPS and treated with PBS, Epirubicin and Aclarubicin and its analysis are available at https://github.com/andrebolerbarros/Chora_etal_2022 (copy archived at swh:1:rev:d86a4302612c313875b919e-3897bec15310b6895) and https://zenodo.org/record/7389633.

The following datasets were generated:

| Author(s) | Year | Dataset title | Dataset URL | Database and Identifier |
| --- | --- | --- | --- | --- |
| Barros A, Pedroso D, Neves-Costa A, Moita LF | 2022 | Murine Bone Marrow Derived Macrophages (BMDM's) stimulated with LPS and treated with PBS, Epirubicin and Aclarubicin | https://github.com/andrebolerbarros/Chora_etal_2022/tree/master/R_Analysis/metadata | Github, Github |
| Barros A, Pedroso D, Neves-Costa A, Moita LF | 2022 | Murine Bone Marrow Derived Macrophages (BMDM's) stimulated with LPS and treated with PBS, Epirubicin and Aclarubicin | https://zenodo.org/record/7389633 | Zenodo, 10.5281/zenodo.7389633 |

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

# Appendix 1

List of primer sequences (mostly retrieved from https://pga.mgh.harvard.edu/primerbank/).

| Mouse primers | | | |
| --- | --- | --- | --- |
| Gene | Forward sequence | Reverse sequence | Primer Bank ID |
| Gapdh | AACTTTGGCATTGTGGAAGG | ACACATTGGGGGTAGGAACA | 6679937a1 |
| IL1b | GAAATGCCACCTTTTGACAGTG | GGCTTGTCACTCGAATTTTGAGA | 118130747c1 |
| Tnf | CTGAACTTCGGGGTGATCGG | GGCTTGTCACTCGAATTTTGAGA | 133892368c2 |
| Cxcl10 | GTGGCATTCAAGGAGTACCTC | TGATGGCCTTCGATTCTGGATT | 10946576a1 |
| IL6 | CCAAGAGGTGAGTGCTTCCC | CTGTTGTTCAGACTCTCTCCCT | 26354667a1 |
| IL10 | GCTGGACAACATACTGCTAACC | ATTTCCGATAAGGCTTGGCAA | 291575143c2 |
| Ccl2 | TTAAAAACCTGGATCGGAACCAA | GCATTAGCTTCAGATTTACGGGT | 6755430a1 |
| Ccl5 | GCGCTCCTGCATCTGCCTCC | CCTTGATGTGGGCACGGGGC | 7305461a1 |
| Nfkb1 | CCTGGATGACTCTTGGGAAA | TCAGCCAGCTGTTTCATGTC | 6679044a1 |
| Nfkb2 | GAACAGCCTTGCATCTAGCC | TCCGAGTCGCTATCAGAGGT | 9506921a1 |
| Nfkbia/Ikba | TGAAGGACGAGGAGTACGAGC | TTCGTGGATGATTGCCAAGTG | 6754840a1 |
| Ifnb1 | TGGGTGGAATGAGACTATTGTTG | CTCCCACGTCAATCTTTCCTC | 6754303c3 |
| Ifna1 | GCCTCGCCCTTTGCTTTACT | CTGTGGGTCTCAGGGAGATCA | Designed by the authors |
| Ifna4 | TGATGAGCTACTACTGGTCAGC | GATCTCTTAGCACAAGGATGGC | 6754294a1 |

