## [Editor Report]

This is an interesting study with several implications for anti-cancer therapy, both systemic and local. This study contributes additional very important cancer cell-specific NF-κB inhibition information which is required for improving anticancer efficacy and reducing systemic toxicity. The conceptual significance is that anthracyclines prevent the induction of pro-inflammatory genes in macrophages without inducing or involving DNA damage response. Confirmation of this mechanism in vivo will help the future development of more efficacious anthracycline drugs specifically optimized for this mechanism without causing major side effects.

---

## [Decision Letter]

[Editors' note: this paper was reviewed by Review Commons.]

---

## [Author Response]

Abbreviations in this reply:

Epirubicin – Epi

Aclarubicin – Acla

Doxorubicin – Doxo

We thank the reviewers and *eLife* for the opportunity to revise and improve our manuscript. All three reviewers agreed that the study has several important implications for anticancer therapy and can contribute to the future development of more efficacious anthracycline drugs with less side effects (DNA damage), therefore reducing their systemic toxicity to potentially improving their anticancer efficacy in patients. The reviewers also stressed that the impact of the findings can extend to other inflammation-associated pathologies, including sepsis, where the new mechanism described may help the development of low-toxicity therapies to suppress inflammation. In line with these observations by the reviewers, our group is a leading participant in a recently initiated clinical trial to assess the potential benefits of Epirubicin in the treatment of patients with severe sepsis and septic shock (ClinicalTrials.gov Identifier: NCT05033808).

Whereas Reviewers #1 and #3 were perfectly satisfied with the depth of the results presented regarding the composition of the NF-κB complexes studied and have not suggested additional experimental work, Reviewer#2 requested more data addressing the possibility that other NF-κB family members may be differently targeted by anthracyclines. Despite being a highly studied transcription factor with well-described oncogenic roles, individual functions of NF-κB subunits, including the particular role of RelA/p65 homodimers, has received very limited attention. In addition, comprehensive cell-specific NF-κB inhibiting information is not available, so we were not able to favor a particular NF-κB family member in our study based on any evidence of possible prominent roles in macrophages. Instead, we chose to study RelA/p65 due to its ubiquity and because “p65 is the transcriptionally active component of the NF-κB species that is most abundant and has the broadest function”, as stated in a comprehensive review dedicated to the DNA binding specificities of each of the NF-κB family members (Wan *et al.*, 2009). To address the concerns of Reviewer#2 we performed a vast series of new experiments, as detailed below. Whereas final and conclusive responses to these longstanding problems, some of which have been elusive for the past four decades, are beyond the scope of our original manuscript, we are confident that our work now allows for a better understanding of the contribution of the NF-κB family member RelA to the gene expression programs induced by inflammatory challenge and, consequently, of the effects of anthracyclines on NF-κB-regulated genes. As Reviewer#2 pointed out, our study opens new avenues for the understanding of the intricate physiological attributes of NF-κB.

Finally, the reviewers unanimously pointed out that the studies were well-conducted and that the interpretations and conclusions were supported by the data displayed; moreover, the reviewers considered that the main outcomes of the experiments performed help not only to explain prior results by others, but also add considerable novelty to the NF-κB field by exposing previously unknown molecular mechanisms.

Point by point:

Reviewer 1:Major comments:The data are convincing as depicted by the in vivo and in vitro panels in the figures.Introduction should be more focused on anti-tumoral drugs and the part related to TLRs and NFkB should be shorted. Moreover, the introduction should end with the aim and not with a declaration of results. As suggestion, this part might joined with the first sentence in the discussion part to better introduce the discussion of findings. References are appropriated.

We have followed the Reviewer’s suggestion and considerably shortened the Introduction; we have also moved the final paragraph into the Discussion and now the Introduction’s final sentence includes only the aim of the work.

As Figures are rich of complex data, especially the in vitro ones, the figure legends need to be better introduced with a brief sentence and a note pointing at what of interest.

Each legend of the main figures now starts with a succinct sentence stressing the most relevant findings and a brief mention to the models and experimental approaches used if relevant.

The conclusion part should be shorter.

The conclusion has been considerably edited and is now much shorter; we believe it has improved considerably and is now easier for the readers to focus on the main findings of the work, as well as the possible applications and clinical impact.

Remove figures from discussion, as this part is only related to the discussion of findings.

Done as suggested.

Minor comments:The “Key resources Table” should be presented as supplemental material and not at beginning of methodology, based on the fact that all reagents and kits have been also described in the text.

A comprehensive key resources table has been includedat the beginning of the Methods section according to the journal’s requirements.

Indeed, the mouse primers table is missing of accession numbers and no amplification protocol as well as the formula used for quantify fold changes have been included.

All accession numbers of the genes (Gene IDs from NCBI) and of the mouse primers (obtained from Primer Bank) can be found in Appendix 1. The protocol for the amplifications and fold change calculations are now included in the methods section .

Moreover, the experimental part needs a better presentation and particularly the declaration of the number of animals used for the in vivo studies and also the number of in vitro tests or replications carried out. More details are necessary for western blot analysis. Please specify the HEK cell line and its specific application in the promoter studies.

We added the following information: 1, the number of animals used in the in vivo experiments has been included in the figure legend; 2, the number of replicates is now stated for each experiment in the corresponding figure legend; and 3, more details are provided in the Methods section for the antibodies used in the WBs, and IF, including the dilutions used. The cells used for the transfections, HEK293, were clearly identified.

Finally, a revision of technical English form should be carried out.

This has been performed by an English native speaker.

Reviewer 2:Major comments:1. Anthracycline-dependent, and DNA damage/ATM-independent, inhibition of inflammatory genes in *E. coli* or LPS stimulated BMDM is generally convincing from dose response studies with sufficient replicates and statistical analyses. However, whether this inhibition in BMDMs (e.g., all the genes shown in Figure 2B) is RelA-dependent or not was not investigated and thus remains unconvincing. LPS stimulation of BMDMs would activate p50/RelA heterodimers and cRel-containing NF-κB complexes but the authors focused only on RelA (RelA homodimer) in the remainder of the investigation. Experimental evidence needs to be provided, such as knockdown or knockout studies, to demonstrate the RelA requirement for anthracycline effects shown in Figure 2B. Moreover, luciferase reporter studies shown in Figure 2F needs to evaluate other NF-κB family members to demonstrate specificity, especially p50+RelA and cRel (+/-p50). If cRel is also affected by anthracycline, the relevance of the rest of the studies, especially NMR studies, focusing on RelA homodimer to the mechanism involved in BMDM becomes highly questionable.

Reviewer#2 pointed out that our work did not clearly establish that the anthracycline-dependent phenotype observed in mouse macrophages is specifically mediated by RelA/p65, namely in limiting cytokine production following pro-inflammatory stimuli. While it is known that Rel proteins are obligate dimers and that LPS is able to activate more than one NF-κB dimer, including RelA/p65 homodimers but also heterodimers formed with other NF-κB factors, the specific contribution of each NF-κB complex to cell responses is not known and has remained elusive for the past decades.

KO studies to test requirement of RelA/p65 for anthracycline effects

In our manuscript, we decided to focus on RelA/p65 because this is a very abundant subunit, ubiquitous, and the most well-studied NF-κB factor. The importance of RelA/p65 is evident in RelA−/− animals, which are embryonic lethal (Geisler *et al.*, 2007). Reviewer#2 requested additional experiments to demonstrate RelA/p65 requirement for the anthracycline effects on transcriptional regulation of pro-inflammatory genes (“Experimental evidence needs to be provided, such as knockdown or knockout studies, to demonstrate the RelA requirement for anthracycline effects shown in Figure 2B.”), and so we decided to perform new experiments. We started by knocking out NF-κB subunits independently in HEK293 cells: RelA/p65, p50 and c-Rel. We did not attempt to perform KOs of these subunits in BMDMs, as we have observed in the past that transfection of these cells, as well as performing viral-mediated nucleic acid delivery, compromises their intrinsic properties, in particular the extent to which pro-inflammatory responses are activated following stimulation by external challenges. In the future we plan to use primary macrophages from KO mouse models instead, which will be the focus of a subsequent manuscript by our laboratory (discussed below). Here, and according to our revision plan, we used CRISPR/Cas9 to knock out the NF-κB factors in HEK293 cells (Author response image 1), which also allowed us to perform reporter gene assays using the same reporter as before to complement the assay in Figure 2F of the manuscript. We did not select individual KO clones after CRISPR/Cas9 because knocking out these subunits is known to compromise cell viability. Therefore, we worked with populations composed of 63 to 75% of KO cells and we were only able to use them immediately after CRISPR/Cas9, as the KO cells tended to die and to be outnumbered by the healthy cells in the population over time (Author response image 1). The protocols and the CRISPR/Cas9 tools are described in the Methods section at the end of this response.

**Author response image 1. sa2fig1:** CRISPR/Cas 9 mediated KOs of NF-κB, subunits in HEK293 cells as described in the methods section of this reply. The resulting populations were tested, at week one post KO, for RNA levels and protein expression of the different subunits. For one round of deletions that we show as an example, we concluded from our and a quantifications that the KO cells in the population were 71% in the case of RelA/p65, 63% in the case of p50 and 75% in the case of c-Rel (A). The percentage of KO cells in the population tended to decrease overtime, as assessed by RNA (B), and for this were only cultured for short periods following gene editing. These KOs were corroborated at the protein level, as shown by WB (C), except for c-Rel, as the antibody that we had available was only able to detect overexpression of the protein (see Author response image 4) and we were never able to detect the much lower levels of the endogenous c-Rel protein.

We then used the KO cell populations to test the effect of anthracyclines on NF-κB-regulated endogenous gene expression. As reported in Figure 2B of the manuscript for BMDMs, endogenous pro-inflammatory gene expression was also downregulated by Epi in CRISPR controls, as shown by TNF mRNA levels (Author response image 2, compare the two blue arrows). In RelA/p65 KO cells, very low levels of TNF mRNA induction were detected (Author response image 2, red arrow), which strongly suggests that RelA/p65 is required for TNF expression and, therefore, for any modulatory effects on that expression, including the effects of anthracyclines. To further substantiate the RelA/p65 requirement for anthracycline effects, we did not detect any decrease in TNF mRNA expression levels in p50 KO or c-Rel KO cells upon stimulation (Author response image 2, light gray arrows), strongly pointing to RelA/p65-containing NF-κB complexes being the main or sole drivers of TNF transcriptional induction, in our experimental conditions. It is likely that a previously described compensatory effect is responsible for the increased RelA/p65 activity in p50 KO and c-Rel KO cells, which leads to increased induction of TNF mRNA in these cells (Author response image 2, light gray arrows). As a final evidence that RelA/p65 is the NF-κB factor mediating anthracycline effects, we observed that Epi was able to downregulate TNF mRNA expression both in p50 KO and in c-Rel KO cells (Author response image 2, dark gray arrows).

**Author response image 2. sa2fig2:** Effects of epirubicin (Epi) on endogeneous TNF expression as assessed by nRNA levels. CRISPR/Cas9-mediated KO populations of HEK293 cells (rel/p65 KO, p50 KO, c-Rel KO, and CRISPR control) which generated as shown in Author response image 1 and were either treated with vehicle or with 2µM of Epi for 4 hours. TNF was added to the medium 1 hour after Epi or cells were left out without any stimulus. mRNA levels of TNF were quantified by qRT-PCR in two independent experiments. One representative experiment is shown with averages and standard deviations of technical replicates. Blue, red and grey arrows are discussed in the text.

We also used the KOs of the NF-κB subunits to test reporter gene expression with the same vector as in Figure 2F of the manuscript. In RelA/p65-deficient cells, the kB-luciferase reporter was not substantially induced by TNF, compared with up to 6x induction of luciferase by TNF in CRISPR control cells (Author response image 3). Because of the lack of reporter expression in RelA/p65 KO cells, it was not possible to test the effects of Epi on reporter gene expression in the RelA/p65 KO cell population. We consider this result to be a strong indication that the kB reporter gene inductions reported in the manuscript are largely due to RelA/p65 and therefore anthracyclines require RelA/p65 for their effects in the luciferase reporter assays. As discussed above, a RelA/p65 compensatory effect was detected in p50 KO cells, leading to luciferase levels slightly above control. On the contrary, in c-Rel KO cells the luciferase signal was not as high as in control cells, which suggests that c-Rel is responsible for a small part of reporter induction, although considerably less prominent than the induction by RelA/p65, the main regulator of TNF-induced luciferase expression in this system.

**Author response image 3. sa2fig3:** Effects of NF-κB subunits on reporter gene expression. HEK293 KO cells (rel/p65 KO, p50 KO, c-Rel KO, and CRISPR control) were stimulated with TNF 4h or 8h and the induction of the luciferase reporter was assessed as fold change over empty vector control.

Specificity assessed by overexpression of NF-κB factors in luciferase reporter assays

The data showed above already started to address the next concern of Reviewer#2, in particular that “luciferase reporter studies shown in Figure 2F need to evaluate other NF-κB family members to demonstrate specificity”. To further investigate this, we co-transfected the NF-κB luciferase reporter (the same vector used in Figure 2F of the manuscript and already used above) together with expression vectors for each of the human full-length NF-κB family members: RelA/p65 (as before, in Figure 2F of the manuscript); p50, and c-Rel (see Methods section at the end of this response). After optimizing the overexpression conditions (Author response image 4 A and B), we tested luciferase induction. We observed that both RelA/p65 alone or double overexpression of RelA/p65 and p50 strongly induced the reporter gene to comparable levels (Author response image 4 and D). As expected, overexpression of p50 homodimers did not induce the luciferase reporter, as this subunit lacks the transactivation domain (TAD) and so it must partner with TAD-containing subunits, such as RelA/p65 or c-Rel, to activate transcription (Figure 4A and C, Hoffmann *et al.*, 2003). We observed that c-Rel overexpression induced transcription only marginally (less than 10x), both alone or in combination with p50 overexpression (Author response image 4 and C). Double overexpression of RelA/p65 and c-Rel was less efficient at transcriptional activation than RelA/p65 overexpression alone (Author response image 4). Our data seems to corroborate previous observations that c-Rel inhibits pro-inflammatory transcription by RelA/p65 (de Jesus *et al.*, 2020)**.** Because c-Rel homodimers are either weak transcriptional activators or repressors in our system, even without any anthracycline treatment, we excluded the possibility that c-Rel is required for the anthracycline effects reported in the manuscript. Levels of protein overexpression of the NF-κB subunits in the conditions used were assessed by WB (Author response image 5 A-C). Importantly, when RelA/p65 was overexpressed simultaneously with p50, a condition that led to strong induction of the reporter gene as discussed above, we observed that both subunits maintained their overexpression at the protein level (Author response image 5).

**Author response image 4. sa2fig4:** NF-κB subunits were overexpressed either alone or two subunits were simultaneously overexpressed and the induction of the luciferase reporter was assessed as fold change over empty vector control. Transfection conditions for reporter and NF-κB subunits were optimised over time (A) and the total amount of transfected DNA was transfected (B) to find the optimal conditions for luciferase signal detection within non-saturating conditions. First, the reporter vector was transfected into HEK293 cells and 24h later the vectors for the overexpressions of the subunits were transacted. Luciferase was read at several time points (eventually we decided to read luciferase signals at 24 hours post transfection, A) and with different amounts of transfected DNA 150 ng of total DNA was chosen as the preferred DNA amount, (B). (C) Both single overexpressions and overexpressions of two subunits simultaneously were performed; overexpression of combinations of two NF-κB subunits aimed at promoting the formation of heterodimers – although we cannot exclude that only homodiners formed in these conditions. (D) We tested luciferase induction after overexpressing combinations of the subunits, D, luciferase induction after the combined overexpression of RelA/p65 and p50 was tested several times in addition to the experiment shown in C; luciferase induction following the double transactions was comparable to the induction by RelA/p65 alone, as shown in three independent experiments, in addition to the experiment in C. Here we show representative assays of at least two independent experiments; I D three independent experiments are shown for comparison. Data refers to averages and standard deviations of technical triplicates.

**Author response image 5. sa2fig5:** NF-κB subunits were overexpressed either alone or two subunits were solved simultaneously over expressed and the protein levels were assessed by WB full stop transfection conditions for reporter and NF-κB subunits were the same as in Author response image 3. (A) Time courses of protein levels were performed for RelA/p65 and for p50 single transactions and compared with empty vector pcDNA. (B) c-Rel overexpression was also detected in cells 24h post transfection, but this antibody showed strong cross reaction with RelA/p65, and therefore limited tests were performed. (C) Simultaneous overexpression of p65 and p50 subunits did not seem to compromise protein levels of any of them.

The specificity of Epi effects was then tested using the luciferase reporter system in the conditions that induced expression of the reporter: RelA/p65 alone or co-transfected with p50. We observed that Epi inhibited luciferase induction in a dose-dependent way when RelA/p65 overexpression was tested alone but also with the double overexpression of RelA/p65 and p50 (Author response image 6); likely, direct RelA/p65 inhibition by Epi compromises reporter expression driven by RelA/p65 homo- and also heterodimers; it is also possible that Epi is able to inhibit p50 directly when in complex with RelA/p65. Nevertheless, we cannot exclude that only RelA/p65 homodimers were inhibited by Epi and that the RelA/p65-p50 heterodimers formed did not suffer from Epi inhibition and were still able to activate transcription; in this case, and taking into consideration the strong effects of Epi towards inhibition of NF-κB targets that we report in the manuscript, RelA/p65-p50 heterodimers could not account for the majority of the NF-κB complexes in cells.

**Author response image 6. sa2fig6:** RelA/p65 was overexpressed either alone or simultaneously with p50 and the induction of the luciferase reporter was assessed as fold induction over empty vector control after incubation with several concentrations of AP from stock transactions were performed as described in the previous Author response images, Epi or vehicle were added 24h post transfection of the subunits and luciferase signal was read at 16h post Epi.

RNA-seq analysis to further explore specificity

From our KO studies and reporter gene assays described above, we could not establish if Epi is able to regulate c-Rel targets or if p50 is also targeted by Epi when in a complex with RelA/p65. To definitely address the question of whether anthracyclines can also regulate targets of NF-κB independently of RelA, we decided to focus on those genes with expression strongly driven by c-Rel or p50 overexpressions, the most likely endogenous targets of c-Rel and p50-containing NF-κB transcriptional complexes. To identify such genes, we performed RNA-seq in HEK293 cells overexpressing the different subunits alone and also in cells with the combined overexpression of RelA/p65 and p50, as this condition also led to strong transcriptional induction (Author response image 4). We found that conditions where RelA/p65 was overexpressed, both alone and in combination with p50, let to the most altered transcriptomic profiles when compared with the transcriptomes of controls (Author response image 7, cluster on the right). This observation aligns with the established notion that RelA/p65 is the NF-κB factor that contributes the most to the transcriptional regulation of NF-κB targets, in line with the results discussed above.

**Author response image 7. sa2fig7:** NF-κB subunits were overexpressed either alone or simultaneously and RNA-seq was performed. Principle Component Analysis (PCA) of the sequencing data was obtained for the following samples: HEK293 cells without any treatment (Cells); TNF stimulated cells (TNF); Epi-treated cells (Epi); TNF-stimulated, Epi-treated cells (EpiTNF); pcDNA (empty vector) -transfected controls (pcDNA); RelA/p65 overexpressing (p65); P50 overexpressing (p50); c-Rel-overexpressing (c-Rel); RelA/p65 on p50 simultaneous transfection (p65p50); Epi-treated pcDNA (empty vector) -transfected controls(EpcDNA); Epi-treated RelA/p65 overexpressing (E65); Epi-treated P50 overexpressing (Ep50); Epi-treated c-rel-overexpressing (c-Rel); and Epi-treated RelA/p65 and P50 simultaneous transfection EP65P50.

A preliminary analysis of the RNA-seq data has uncovered targets of RelA/p65, p50 and c-Rel overexpressions, as shown in the volcano plots of each single overexpression (Author response image 8). We confirmed that NF-κB factors are mostly of gene inducers: 3137 upregulated targets in RelA/p65 overexpression; 1128 upregulated in p50 overexpression; and 1241 upregulate in cRel overexpression. Downregulated genes, either directly or indirectly, were the minority: 416 downregulated targets in RelA/p65 overexpression; 282 downregulated in p50 overexpression; and 570 downregulate in cRel overexpression. The top 50 upregulated genes for each of the overexpressions showed the expected RelA/p65 targets genes and also putative targets of p50 and c-Rel (Author response image 9). We have analysed the first 200 upregulated genes in the RelA/p65 overexpression and asked which were the most represented GO TERMS regarding the biological process (BP). For that we used the Database for Annotation, Visualization and Integrated Discovery (DAVID) as before in the manuscript (https://david.ncifcrf.gov). The terms are all related to immune responses, as for example chemokine and cytokine signalling and secretion and, which is in line with the known roles for RelA/p65, therefore further validating the relevance of the RNA-seq data (Author response image 9).

**Author response image 8. sa2fig8:** Volcano plots of the RNA-seq data for the overexpressions of RelA/p65, p50 and cRel. Volcano plots of the conditions: (A) overexpression of RelA/p65 vs pcDNA; (B) overexpression of p50 vs pcDNA; and (C) overexpression of p50 vs pcDNA. Geenes with log2FoldChange expression above 2 or below -2 were considered to be upregulated or downregulated.

**Author response image 9. sa2fig9:** Most induced genes from the RNA-seq following overexpression of the NF-κB subunits and GO TERMS of the most significantly upregulated RelA/p65 targets. I list of the 50 most significantly upregulated genes from the RNA-seq for each of the subunits overexpressed when compared with this expression in control P DNA; The lists start with the most significantly induced mRNA in the top of each column the top 200 upregulated mRNAs in RelA/p65 overexpression where analysed with the use of DAVID (https://david.ncifcrf.gov) and the 10 most represented GO TERMS for biological processes (GOTERM_BP) or shown from the most significant at the top.

We are now in the process of compiling a comprehensive list of p50 and c-Rel targets with the help of the Bioinformatics Unit at IGC. This work in progress also aims to discriminate which genes are upregulated by these subunits but are not found to be targets of RelA/p65. A list of genes induced by p50 or by c-Rel but not by RelA/p65 is of particular interest to us, as we intend to thoroughly test whether these genes are also targeted by the anthracyclines. Our ongoing bioinformatics analysis is also exploring the effects of Epi-treatments on p50 and cRel targets, as Epi-treated conditions were included in the RNA-seq experiment. Future results from the genes selected will be similar to the analysis already performed for TNF (Author response image 10): we will be able to compare the expression of each selected p50 and c-Rel target in conditions with and without Epi to see if anthracycline treatments also dowregulate the expression of those genes. Our analysis will then provide a more definitive picture of the extent to which anthracyclines regulate NF-κB targets. Our laboratory will dedicate considerable efforts to these studies, which will only be presented in the future because of the thorough analysis and validation that is required. The exhaustive identification and validation of the NF-κB genes regulated by Epi, with the possibility of recognizing the preferred composition of each NF-κB complex that is being targeted by the anthracyclines for each gene, will considerably expand our knowledge of NF-κB biology, but also of the particularities of locus-specific drug targeting, in particular in inflammation. We consider this to be a very relevant and poorly understood aspect, as anthracyclines are already known to target discrete chromosomal regions in cancer cells to induce histone eviction, and it would be fundamental to explore whether anthracycline recruitment to particular chromatin domains could eventually be related to the presence of distinct NF-κB dimers (references in the manuscript: Pang *et al.*, 2013 and Yang *et al.*, 2013).

**Author response image 10. sa2fig10:** RNA-seq data for the expression of TNF in the different conditions tested, shown as an example of Epirubicin (Epi)-regulated NF-κB target gene. The mRNA levels corresponding to TNF expression, as obtained by the RNA-seq. Controls were untransfected cells: HEK293 cells without any treatment (Cells); TNF-stimulated cells (TNF); Epi-treated cells (Epi); TNF-stimulated, Epi-treated cells (EpiTNF); HEK293 sounds where transfected and treated as follows: pcDNA (empty vector) -transfected controls (pcDNA); RelA/p65 overexpressing (p65); P50 overexpressing (p50); c-Rel-overexpressing (c-Rel); RelA/p65 and p50 simultaneous transfection P65P50; epi-treated pcDNA (empty vector) -transfected controls (EpcDNA); epi-treated RelA/p65 overexpressing (Ep65); Epi-treated P50 overexpressing (Ep50); Epi-treated c-Rel-overexpressing (EcRel); and Epi-treated RelA/p65 and P50 simultaneous transfection (Ep65p50). Values are arbitrary and represent fold induction over the condition “cells” alone, for which basal mRNA levels were considered to be zero.

Summary

We are confident that the findings shown in this section of the reply to Reviewer#2 provided information that is novel and hopefully contributed to clarify the doubts raised. Our results allowed us to establish that RelA/p65 is the main NF-κB factor responsible for the effects of the anthracyclines (Epi) in the cell models tested, both in the regulation of endogenous gene expression, as well as in regulating the kB reporter. For the relevance of these observations, we decided to include a new supplementary figure in the manuscript, Supp. Figure 4, that contains part of the information provided here. We only included data concerning RelA/p65 KO and no data related to p50 or c-Rel because RelA/p65 is the focus of our manuscript and the sole NF-κB factor that we studied structurally, by IF, and in reporter gene assays. Another change we introduced in the manuscript was in Figure 2: the former 2E part of the panel, which showed redundancy with 2F, was replaced by data stressing that TNF gene is regulated by Epi at the level of transcription and not at the level of stability of its mRNA, which to us further suggests that anthracyclines regulate transcription of NF-κB targets.

We are aware of the limitations of using a cell line such as HEK293 and reporter gene assays, and we acknowledge that we did not address the dispute in macrophages. Nevertheless, and because the experimental approaches used were chosen to mimic inflammatory stimulation and the induction of pro-inflammatory transcriptional programs, we believe that the findings shown here are relevant for the understanding of the effects of anthracyclines in inflammation. In the future, we plan to use genetically modified mouse models and isolate macrophages from those animals to do in vivo and *ex-vivo* studies. RelA/p65 deletion is embryonically lethal (Geisler *et al.*, 2007) and therefore RelA/p65 KO cells can only be obtained in inducible systems. Adult RelA/p65 inactivation is possible in RelA/p65 flox/flox mice using Cre/loxP approaches, either for whole-body recombination and deletion, or using a Cre driver such as LysM-Cre to achieve deletion of RelA/p65 only in macrophages. In these KO animals we will also be able to test the protective in vivo effects by anthracyclines using the models of sepsis and septic shock which are well established in our laboratory, as we did in the past (reference in the manuscript: Figueiredo *et al.*, 2013). The work presented in the manuscript and in this response explored anthracycline interactions with RelA/p65 and showed that these interactions are likely to limit the extent of pro-inflammatory responses, which may have beneficial consequences in conditions associated with exacerbated inflammation. A model of target gene regulation by Epi is shown in Supp. Figure 12 of the manuscript.

2. A related significant concern is the use of the "kB-33 promoter sequence" (page 10 and thereafter) in the NMR study (Figures 4 and 5 and associated supplemental data). The description of this NF-κB binding sequence was insufficient in the manuscript. In fact, this sequence is not even a "promoter" sequence as stated. This is an artificial sequence derived from a duplication of the 3' half of the immunoglobulin kappa kB enhancer element (Thanos & Maniatis, Mol. Cell. Biol. 15: 152-164, 1995), which was cited in the Chen et al. 1998 paper which was in turn cited by the authors. This sequence selectively binds RelA homodimers and not p50/RelA heterodimers or p50 homodimers. Given that the role of p65 homodimers in regulating inflammatory genes in BMDM is not established (in the current study or by others), the use of this artificial binding site in the NMR study and the relevance of the results obtained to the mechanism of action of anthracycline in repressing NF-κB dependent target genes in BMDM are highly questionable. If, on the other hand, the authors are able to demonstrate the requirement of RelA homodimers, but not p50/RelA, p50/cRel heterodimers, or cRel homodimers, for gene regulation shown in Figure 2B, this could help explain why certain other NF-κB target genes, such as IFN β or survival genes, are not affected by anthracyclines as these targets are known to be regulated by p50/RelA heterodimers (e.g., IFN β gene described in Thanos & Maniatis cited above).

We confirm that we have specifically chosen a sequence that selectively binds RelA homodimers and not p50/RelA heterodimers or p50 homodimers because we only assessed direct transcriptional inhibition (luciferase assay) by RelA and not by the other NF-κB subunits in the original manuscript; while we think it would be of great interest to extend the NMR studies to other subunits, we believe that the NMR studies that we performed with RelA already provide considerable information regarding the possible molecular mechanisms that explain the inhibition of RelA homodimers that are capable of driving the transcription of luciferase with a kB promoter. Instead, we have decided to concentrate our efforts and resources on testing a wide variety of conditions affecting the binding that may prove relevant in physiological conditions. The conditions tested included a range of doses of the anthracyclines, different possible stoichiometries between the components, alternative orders of addition of the players of the system, and very importantly, three members of the anthracycline family that each have distinctive biological features. We agree that it will be helpful, in the future, to expand the structural analysis to other sequences, not only targeted by other NF-κB subunits, but also by other transcription factors with known roles in regulating inflammation, such as the IRFs and STATs families for example. However, that extends beyond the scope of this work and due to the nature of the experimental approaches required to generate reliable structural data, will only be possible in the future.

3. By demonstrating the significant reduction of serum TNF levels in mice exposed to *E. coli* + Epi or Acla relative to *E. coli* group alone, the authors concluded that "our findings have in vivo physiological relevance" (page 6). However, the difference of TNF levels is ~1250 pg/ml to ~800 pg/ml, only ~36% reduction. It is unclear whether this reduction would have "physiological relevance". Thus, the authors should measure the physiological outcome of septic shock, such as death, to determine whether anthracyclines have significant physiological impact or not in vivo. This is important given the authors claim that the mechanism being studied have implications in cancer and sepsis therapies in patients.The additional experiments suggested above are within the capacity of the authors, not very time consuming and not expensive relative to what have already being performed and therefore reasonable to make the study and the claim more conclusive and significant.

Because of the reduction of circulating TNF levels in the sepsis mouse model after anthracycline treatment, the reviewer suggested the measurement of a physiological outcome, more specifically the kinetics of death. Indeed, the physiological relevance of anthracyclines in sepsis was the central focus of a previous study conducted by our laboratory, published in Immunity in 2013 (reference in the manuscript: Figueiredo *et al.*, 2013). In that publication, we compared the survival of WT animals subjected to sepsis (using the polymicrobial infection induced by cecal ligation and puncture model, CLP) and treated with control (PBS) or treated with anthracyclines. Figures 1A and 1B in that publication showed that the anthracyclines Epirubicin, Doxorubicin and Daunorubicin increased the survival of the mice subjected to CLP by up to 80%, without the use of antibiotics. We agree with the reviewer that the in vivo results have considerable impact for the development of future treatments targeting cancer and sepsis. The previously published observations that anthracyclines are protective in the mouse model of sepsis are a strong indication that anthracyclines will play critical roles in sepsis therapies in patients. This is being tested in a clinical trial conducted in Germany and with the participation of the authors (ClinicalTrials.gov Identifier: NCT05033808). More specifically, the trial will compare low doses of Epirubicin with placebo for the treatment of sepsis and septic shock. We see no need to repeat or extend the in vivo tests to additional anthracyclines, for which we would need a substantial number of animals and this would not be ethically appropriate.

Minor comments:1. In addition to the insufficient/inaccurate description of the kB-33 promoter sequence above, the description of the kB-Luc reporter used in the method section ["the NF-κB firefly luciferase reporter construct (kB-luc),.. were previously described (Anrather et al., 1999)"] (page 22) require a better description. The cited paper (Anrather et al. 1999) does not have the direct information on the reporter construct and one has to look up a cite paper within the reference (i.e., Brostjan C et al. J. Immunol. 158: 3836-3844, 1997) to find the relevant reporter vector.

The description has been considerably expanded in the Methods section: we have now included the sequences of the two kB sites in the vector; in addition, other sequences in the vector such as the promoter and enhancer have been described. Finally, there is a new reference to the originally identified consensus sequence for comparison.

2. In vivo sepsis study in the result section refers to "we co-administered Epi or Acla to mice that were challenged with LPS" (page 6). However, the methods and figure legend indicate the use of *E. coli*.

We thank the Reviewer for calling our the attention to this. We have now changed to “fixed *E coli*,” the model that was used to assess the relevance of cytokine modulation by anthracyclines in vivo.

3. The authors uses the term "inflammation" multiple times in the text to describe the induction of inflammatory genes in BMDM, but this term is misleading because inflammation refers to a physiological reaction in vivo. The authors should consider the use of more accurate description.

We agree with the Reviewer that a more accurate description was required in some of the sentences. We have now replaced a few of the mentions to “inflammation” by “induction of gene expression”, or “induction of pro-inflammatory gene expression” or “pro-inflammatory transcriptional programs” to specifically reflect our observations that RelA is capable of reducing the transcription of genes that take part in the inflammatory responses in macrophages.

4. All references lack volume and page numbers. This needs to be fixed.

This has been corrected.

5. Figure 1A: Atm-/- cells show statistically significantly different levels (although the difference is small in magnitude) relative to WT groups for some Epi doses but these differences were largely ignored in the manuscript.

We have now acknowledged these small differences in the text, more specifically in the section “Cytokine secretion and DNA damage”. Although the data clearly shows that ATM is not required for the RelA-mediated reduction of gene expression of pro-inflammatory genes, we are aware that inflammation progresses differently in ATM-/- animals compared with WT animals (shown for example in Rodier *et al.*, 2009), and therefore direct comparisons should be avoided (this is part of the reason why we favor the “fold change” analysis for cytokine induction results). In addition, because anthracyclines target DNA, and ATM-/- cells are considerably defective in DNA damage repair and are more prone to accumulate DNA breaks, we do not exclude that Epirubicin may interact differently with the ATM-/- chromatin when compared with the WT chromatin.

6. Many of the data shown in supplemental figures lacked statistical analysis.

The statistical analysis in this manuscript was performed for all data plotted in graphs, whether the data was presented in the main figures or as supplementary material. In fact, all graphs in the main figures display the results of those statistical analyses visually. However, for the supplementary figures, we did not thoroughly display the results of the statistical tests, which was rightfully pointed by the Reviewer. What we have now decided to do was to mention the statistical analysis in the figure legends of all supplementary figures and to reference the results of the tests. We believe that the figures are now more informative and we thank the reviewer for suggesting this improvement.

Reviewer 3:My comments/suggestions:1. Although ATM independence in repressive effects on transcription of pro-inflammatory cytokines is convincingly demonstrated, how likely that ATR could be involved.

The roles of the ATR factor (ataxia telangiectasia and Rad3-related protein) in inflammation are also a matter of great interest to us and we have dedicated intense efforts to study this particular DNA damage factor in macrophages and other cells. Importantly, whereas ATM and ATR share roles in double-strand DNA damage, these two master kinase activities contribute distinctly to the resolution of other types of cellular injuries, such as replication stress (recently reviewed in Williams *et al.*, 2021). We have collected data but we would not like to share those preliminary results at this time and expect that the editors and reviewers keep this piece of information under strict confidentiality. For this, we thank the editors and reviewers in advance.

2. Please indicate scale bar in figure 3A.

This has been included.

3. In Figure S1 E and F, what are the dose and times of treatments.

The details of doses and kinetics have been provided.

4. There are no description of Figure S2E-I panels in results.

We thank the Reviewer for calling the attention to this and have included the specific reference to these panels in the text.

5. Figure 3 legend, please provide times of *E. coli* treatments in A, and Dose of MG132 in C.

The details of doses and kinetics have been provided.

6. Please provide Figure S1G as the legend states panel G description.

Reference to panel G has been removed from the legend of figure S1.

Methods:

KO studies

We performed CRISPR/Cas9 mediated KOs of NF-κB subunits in HEK293 cells and for that we used commercially available double nickase plasmids (Santa Cruz, Author response table 1) consisting of a pair of plasmids each encoding a D10A mutated Cas9 nuclease and a target-specific 20 nt guide RNA (gRNA). Plasmid transfection was performed as described in the manuscript using the amounts of DNA suggested by Santa Cruz. We chose to use nickases because they generate single-strand rather than double-strand breaks, so when used with two adjacent gRNA, the probability of off-target editing is lower. The resulting population was tested (at week 1 of KO) for RNA levels and protein expression of the different subunits.

**Author response table 1. sa2table1:** Tools for CRISPR/Cas9 in HEK293 cells.

Human Gene	Protein	Double Nickase Plasmid
RELA	p65 (RelA) subunit	sc-400004-NIC-2
NFKB1	P50 subunit	sc-400087-NIC-2
REL	c-Rel	sc-400478-NIC-2
Scrambled control RNA sequence	-	sc-437281

RNA

RNA extraction, cDNA synthesis and qRT-PCR were done as explained in the main Methods section of the manuscript. For one round of deletions that we show as an example, we concluded from RNA quantifications that the KO cells in the population were 71% in the case of RelA/p65, 63% in the case of p50 and 75% in the case of c-Rel (Author response image1). The percentage of KO cells in the population tended to decrease over time as assessed by RNA levels (Author response image 1) and therefore we were careful to only culture cells for short periods following gene editing. These KOs were corroborated at the protein level, as shown by WB. The sequences of the primers were retrieved from Primer Bank (https://pga.mgh.harvard.edu/primerbank/) and can be found in Author response table 2.

**Author response table 2. sa2table2:** Human primers used.

Human Gene	Forward Primer	Reverse Primer
RELA/ p65	ATGTGGAGATCATTGAGCAGC	CCTGGTCCTGTGTAGCCATT
NFKB1/ p50	GAAGCACGAATGACAGAGGC	GCTTGGCGGATTAGCTCTTTT
REL/ c-Rel	CAACCGAACATACCCTTCTATCC	TCTGCTTCATAGTAGCCGTCT
GAPDH	GAGTCAACGGATTTGGTCGT	TTGATTTTGGAGGGATCTCG

Antibodies

The KOs were also corroborated at the protein level, as shown by WB. The antibodies used in this reply were from Santa Cruz: anti p65/RelA antibody (C-20) sc-372 at 1:2000 dilution; anti p50 antibody (NLS) sc-114 at 1:2000 dilution; and anti c-Rel antibody (C-terminus) sc-71 at 1:1000 dilution.

Overexpressions

The full-length RelA overexpression construct, which was previously used in the manuscript, is a pcDNA3-based vector and was described in the Methods section (reference in the manuscript: Anrather et al., 1999.). Full-length p50 and c-Rel-containing overexpression constructs were created similarly.

RNA-seq

RNA integrity was assessed in a Fragment Analyser (Agilent Technologies), and samples with RNA Quality Number (RQN)>8 were further used for mRNA-library preparation using SMART-Seq2. Illumina libraries were generated with the Nextera based protocol and libraries quality were assessed in Fragment Analyzer before sequencing. Sequencing was carried out in NextSeq 2000 Sequencer (Illumina) at the IGC Genomics facility using 100 SE P2 kit and 25 million reads per library. Sequencing data was demultiplex and converted to FASTQ format using bcl2fastq v2.19.1.403 (Illumina).

References:

de Jesús TJ, Ramakrishnan P. NF-κB c-Rel Dictates the Inflammatory Threshold by Acting as a Transcriptional Repressor. iScience. 2020 Mar 27;23(3):100876. doi: 10.1016/j.isci.2020.100876.

Geisler F, Algül H, Paxian S, Schmid RM. Genetic inactivation of RelA/p65 sensitizes adult mouse hepatocytes to TNF-induced apoptosis in vivo and in vitro. Gastroenterology. 2007 Jun;132(7):2489-503. doi: 10.1053/j.gastro.2007.03.033.

Hoffmann A, Leung TH, Baltimore D. Genetic analysis of NF-kappaB/Rel transcription factors defines functional specificities. EMBO J. 2003 Oct 15;22(20):5530-9. doi: 10.1093/emboj/cdg534.

Leung TH, Hoffmann A, Baltimore D. One nucleotide in a kappaB site can determine cofactor specificity for NF-kappaB dimers. Cell. 2004 Aug 20;118(4):453-64. doi: 10.1016/j.cell.2004.08.007.

Martin EW, Chakraborty S, Presman DM, Tomassoni Ardori F, Oh KS, Kaileh M, Tessarollo L, Sung MH. Assaying Homodimers of NF-κB in Live Single Cells. Front Immunol. 2019 Nov 7;10:2609. doi: 10.3389/fimmu.2019.02609.

Oeckinghaus A, Ghosh S. The NF-kappaB family of transcription factors and its regulation. Cold Spring Harb Perspect Biol. 2009 Oct;1(4):a000034. doi: 10.1101/cshperspect.a000034.

Tsui R, Kearns JD, Lynch C, Vu D, Ngo KA, Basak S, Ghosh G, Hoffmann A. IκBβ enhances the generation of the low-affinity NFκB/RelA homodimer. Nat Commun. 2015 May 7;6:7068. doi: 10.1038/ncomms8068.

Rodier F, Coppé JP, Patil CK, Hoeijmakers WA, Muñoz DP, Raza SR, Freund A, Campeau E, Davalos AR, Campisi J. Persistent DNA damage signalling triggers senescence-associated inflammatory cytokine secretion. Nat Cell Biol. 2009 Aug;11(8):973-9. doi: 10.1038/ncb1909.

Wan F, Lenardo MJ. Specification of DNA binding activity of NF-kappaB proteins. Cold Spring Harb Perspect Biol. 2009 Oct;1(4):a000067. doi: 10.1101/cshperspect.a000067.

Williams RM, Zhang X. Roles of ATM and ATR in DNA double strand breaks and replication stress. Prog Biophys Mol Biol. 2021 May;161:27-38. doi: 10.1016/j.pbiomolbio.2020.11.005.